# Fine-grained Analysis of Brain-LLM Alignment through Input Attribution

## Abstract

Understanding the alignment between large language models (LLMs) and human brain activity can reveal computational principles underlying language processing. ~~We introduce a fine-grained input attribution method to~~ This work describes a pipeline to apply attribution methods to the brain-LLM alignment setting to identify the specific words most important for ~~brain-LLM alignment, and~~ this alignment. As a case study, we leverage it to study a contentious research question about brain-LLM alignment: the relationship between brain alignment (BA) and next-word prediction (NWP). ~~Our findings reveal~~ Across two naturalistic fMRI datasets, we find that BA and NWP rely on largely distinct word subsets: NWP exhibits recency and primacy biases with a focus on syntax, while BA prioritizes semantic and discourse-level information with a more targeted recency effect. This work advances our understanding of how LLMs relate to human language processing and highlights differences in feature reliance between BA and NWP. Beyond this study, our attribution method can be broadly applied to explore the cognitive relevance of model predictions in diverse language processing tasks.

## 1 Introduction

An increasing number of studies investigate the similarities between pretrained large language models (LLMs) and the human brain, showing a significant alignment between brain activity patterns and LLM activations when both are exposed to the same linguistic input Toneva & Wehbe (2019); Schrimpf et al. (2021); Goldstein et al. (2022). According to previous work (Aw & Toneva, 2023; Goldstein et al., 2024; Zhou et al., 2024), we define brain alignment (BA) as the performance of brain encoding models that predict brain activity given LLM representations, typically measured as the Pearson correlation between true and predicted activity. Understanding the reasons for this alignment has the potential to provide new insights into the computational principles of the human brain and, in turn, suggest ways to improve current LLMs. Typical approaches for studying reasons for brain-LLM alignment either perturb model representations (Toneva et al., 2022; Oota et al., 2023b; 2024), or the input itself (e.g. permuting the order of words in particular ways) ~~Merlin & Toneva (2024); Kauf et al. (2023)~~(Merlin & Toneva, 2024; Kauf et al., 2023). While informative, these methods can limit the ability to interpret how alignment arises from the input natural stimulus.

In this work, we ~~develop a more fine-grained approach for interpreting the reasons for brain-LLM alignment: an end-to-end input attribution approach which estimates the importance of individual words in the input of an LLMfor its BA (illustrated in~~ introduce a pipeline that integrates established attribution methods into the BA setting, allowing us to estimate how individual input words contribute to an LLM's BA predictions (Figure 1). Specifically, we use gradient-based attribution methods to identify the words that are important for a specific LLM to accurately predict brain activity elicited by the same language input. ~~Thanks to our framework~~In this way, we can ~~get insights into what contextual information is relevant to achieve the measured BA: e.g., where are important words placed in the input context? Does the LLM focus on recent words or words that are further away in the past? Do important words for brain-LLM alignment mostly represent semantic/syntactic/discourse features?~~ directly characterize how and where an LLM functionally draws information from the input when predicting brain activity: for example, identifying where important words are located in the context window. This enables us to quantify positional biases, i.e., whether the model relies primarily on recent tokens or integrates more distributed information

over longer spans, and allows us to determine which linguistic properties (semantic, syntactic, or discourse-related) contribute most strongly to alignment. These analyses allow us to move beyond coarse perturbations and obtain a fine-grained map of the contextual features that give rise to measured BA.

We further showcase how this input attribution ~~approach~~ pipeline can be applied to investigate a contentious research question about brain-LLM alignment: the relationship between BA and next-word prediction (NWP). Specifically, while some works have shown a strong relationship between the two ~~(Schrimpf et al., 2021; Goldstein et al., 2022)~~ (Schrimpf et al., 2021; Goldstein et al., 2022; Hosseini et al., 2024), other works have demonstrated additional important factors for the alignment of LLMs to the brain, such as syntactic information (Oota et al., 2023b) and specific semantic information (Merlin & Toneva, 2024). To investigate the relationship between BA and NWP, we contrast the attributions for brain-LLM alignment with those obtained from the same models during prediction of the next word for 5 open-source pretrained LLMs. ~~Such comparison highlights differences in what each task draws from the same representation extracted from a frozen LLM. According to previous work~~ Across two naturalistic fMRI datasets, this comparison highlights that, even though both tasks operate on the same underlying representations, they functionally draw on different parts of the input to perform their respective objectives. Consistent with prior observations (Merlin & Toneva, 2024), we ~~demonstrate that BA relies on important input cues that are overlooked in NWP , such as specific semantic information~~ find that BA gives functional weight to semantic information that NWP often overlooks, not because NWP lacks semantics, but because its objective prioritizes short-range syntactic predictors of the next token. Additionally, ~~we provide further insights into how much information is used by each task~~ word-level attributions reveal how broadly each task draws information from context (attribution spread) ~~, and where it is placed in the context (recency/primacy biases).~~ and expose distinct positional profiles: NWP exhibits sharp primacy/recency peaks, whereas BA integrates information more broadly across the context.

~~Across LLMs, we find that BA and NWP rely on largely distinct subsets of input words. We show that the attribution spread over context words is higher for NWP at early layers and for BA in middle and late layers, suggesting that BA more strongly relies on higher-level, semantically-richer representations. Additionally, while NWP consistently displays strong recency and primacy biases, BA shows a more focused recency pattern. Further analyses reveal that NWP emphasizes syntactic features, while BA draws more heavily on semantic and discourse-level information .~~ Our main contributions can be summarized as follows:

- We ~~develop a novel end-to-end attribution approach for~~ present a pipeline to apply attribution methods to brain-LLM alignment ~~to enable fine-grained interpretability analyses~~ and analyze the resulting attribution patterns.
- We find that transformers, state-space models (SSMs), and hybrid architectures behave largely similarly in terms of BA.
- We present a case study of how our approach can be used to investigate a contentious question in brain-LLM alignment: the relationship between BA and NWP.
- We show that BA and NWP rely on different input features and context integration patterns: NWP shows recency and primacy biases and strongly relies on syntactic information, while BA has a more pronounced recency bias and also attributes high importance to semantic and discourse-level cues.

## 2 RELATED WORK

A growing body of research has investigated the alignment between brain activity during language comprehension and LLM representations, showing shared structure across biological and computational language systems Wehbe et al. (2014b); Jain & Huth (2018a;b); Toneva & Wehbe (2019). ~~Goldstein et al. (2022)~~ Schrimpf et al. (2021), Goldstein et al. (2022), and Hosseini et al. (2024) suggested that NWP is a major driver of this alignment, with higher predictive performance correlating with stronger BA. However, more recent studies argue that predictive accuracy does not fully explain brain–LLM similarity: NWP is mostly driven by local, word-level cues, such as short-range syntactic dependencies (Oh & Schuler, 2023), while BA also depends on a model's ability to

encode higher-order linguistic features such as syntax and semantics (Oota et al., 2023b; Merlin & Toneva, 2024). Recent evidence further shows that BA and NWP correlate strongly only early in training, then decouple as models acquire more abstract capabilities such as reasoning and world knowledge (AlKhamissi et al., 2025). Our work is complementary: rather than comparing raw alignment scores, we extend word-level attribution analysis to BA, thus enabling direct comparison between the information that drives BA, NWP, or both.

Our work is also related to research on positional biases in LLMs. Prior attribution-based work reported strong recency effects (i.e., high attributions on the most recent tokens) in transformers and SSMs (Oh & Schuler, 2023; Wang et al., 2025), while primacy (i.e., high attribution on early-position tokens) was found only through retrieval-based tasks (Liu et al., 2024; Morita, 2025). Using a unified attribution framework, we show that both biases are present in NWP across architectures, whereas BA exhibits a more pronounced but broader recency profile.

Attribution has also been applied to brain–LLM alignment, but with different aims: Russo et al. (2022); Rahimi et al. (2025) used NWP attributions as the input features used to predict brain activity with the goal of improving alignment scores (i.e. preditive performance). In contrast, we predict brain activity using LLM layer embeddings and then explain this prediction using our end-to-end attribution framework. By comparing attribution overlap, spread, positional patterns, and linguistic feature composition between BA and NWP, our framework provides new insights into the nature of brain–LLM alignment and reveals complementary representational demands of next-word versus brain-activity prediction.

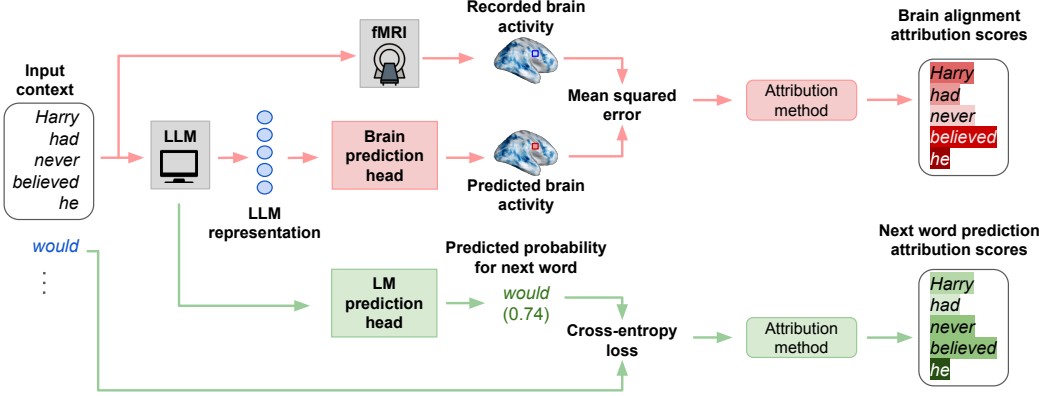

Figure 1: Overview of the attribution pipeline for BA and NWP. The BA pipeline (pink) extracts token embeddings for the input contexts through a pretrained LLM, aggregates them into TR-level embeddings, that are passed to the brain prediction head. Attribution scores are computed with respect to the mean squared error between predicted and recorded brain responses. The NWP pipeline (green) uses the same input to predict the subsequent word via the model's language modeling head, with attribution scores computed based on the cross-entropy loss. In both cases, token-level attributions are aggregated to produce word-level importance scores, allowing for direct comparison of the linguistic features driving each task.

## 3 METHODS

We develop an end-to-end attribution framework to identify the words in the input that are important for the LLM to correctly predict brain activity. By comparing attribution scores obtained for BA and NWP, we can then better understand how LLMs represent information relevant to brain-activity versus next-word prediction. In this section, we outline the data, models, and computational framework used to investigate the differences between BA and NWP through input attribution.

### 3.1 BRAIN DATA

We use a publicly available fMRI dataset (Wehbe et al., 2014a) containing brain recordings of 8 subjects reading chapter 9 of *Harry Potter and the Sorcerer's Stone* (HP) (Rowling et al., 1998).

Table 1: LLMs used in our study, along with their most relevant characteristics.

| Name | Hidden size | Context length | # Layers | Model family |
|------|------------|----------------|----------|--------------|
| Falcon3-1B Team (2024) | 2048 | $4K$ | 18 | Transformer |
| Gemma-2B Mesnard et al. (2024) | 2048 | $8K$ | 18 | Transformer |
| Llama3.2-1B Meta AI (2024) | 2048 | $128K$ | 16 | Transformer |
| Mamba-1.4B Gu & Dao (2023) | 2048 | $2K$ | 48 | SSM |
| Zamba2-1.2B Glorioso et al. (2024) | 2048 | $4K$ | 38 | Hybrid |

The chapter is composed of $N = 5176$ words, presented to the subjects one-by-one for a duration of 0.5 seconds each. The fMRI recordings are sampled at a fixed repetition time (TR) of 2 seconds. The chapter was divided into four runs of similar length, with a short break between runs. We use this dataset primarily because it is one of the public fMRI dataset with the most data per participant, making it well-suited for estimating BA and therefore widely used in prior studies on brain–LLM alignment (Toneva & Wehbe, 2019; Aw & Toneva, 2023; Merlin & Toneva, 2024). However, to test the generalizability of our findings, we perform experiments on an additional dataset, the Moth Radio Hour (MRH) (Deniz et al., 2019), and report results in Appendix D. Beyond the analyses of attribution spread and positional patters, the HP Harry Potter dataset also offers word-level annotations for high-level linguistic features, spanning semantic, syntactic, and discourse categories. We leverage these annotations to conduct a complementary analysis focused on the HP Harry Potter dataset, allowing us to examine which types of linguistic content are prioritized by BA and NWP (see Section 4). We provide more details about the MRH Moth Radio Hour dataset and on the HP Harry Potter's word-level annotations in Appendix A.1

### 3.2 Language models

We analyze five pretrained LLMs with 1 to 2B parameters, sourced from HuggingFace (Wolf et al., 2019). This parameter scale offers a balance between computational feasibility and representational power, allowing us to conduct large-scale attribution analyses while preserving brain-relevant linguistic capabilities. Our selection includes three transformers, one SSM, and one hybrid architecture. We include SSMs because they are explicitly designed for efficient long-context processing (Gu et al., 2021), a property that may be especially relevant for modeling the extended temporal structure of natural language and aligning with brain dynamics. This architectural diversity enables us to explore how attribution and BA differ not only across tasks and layers, but also across model classes. Key architectural features are summarized in Table 1, and further details, including training data specifications, are provided in Appendix A.2.

### 3.3 Input attribution

To interpret which input words most influence BA and NWP, we compute input attributions using gradient-based methods implemented via the Captum library (Kokhlikyan et al., 2020). These methods are model-agnostic, efficient, and well-suited for large-scale comparisons across models and layers. In contrast to perturbation-based techniques, gradient-based approaches are significantly faster and more broadly applicable, as they do not rely on model-specific backward rules, such as those required by LRP (Montavon et al., 2019). Among gradient-based methods, we primarily use Gradient × Input (GXI) (Shrikumar et al., 2016), due to its favorable balance between computational efficiency and interpretability. To validate the robustness of our results, we additionally apply Integrated Gradients (IG) (Sundararajan et al., 2017) to two representative models perform analyses using Integrated Gradients (Sundararajan et al., 2017) and SmoothGrad (Smilkov et al., 2017), obtaining qualitatively similar results to GXI. While IG provides a more theoretically grounded estimate by integrating gradients along a path from a baseline to the actual input, it is significantly more computationally intensive and therefore not feasible for large-scale analyses. Gradient × Input. Formal definitions and implementation details for both GXI and IG all used attribution methods are provided in Appendix A.3, while full results for Integrated Gradients and SmoothGrad are presented in Appendix E.

### 3.3.1 END-TO-END INPUT ATTRIBUTION FOR BRAIN ALIGNMENT

Our goal is to compute input attributions that identify which words in the input context are most important for predicting brain activity. To this end, we develop a gradient-based attribution framework that integrates (1) contextual word representations from a language model, (2) a trained brain encoding model, and (3) a differentiable projection layer that enables gradient flow from brain predictions back to the input. This framework allows us to assign word-level importance scores with respect to the model's ability to align with neural data.

**Brain activity prediction.** We begin by extracting contextual embeddings from a pretrained LLM. For each word $w$ in the text, we construct a context of $L = 640$ words with $w$ as the final word. These contexts are fed into a pretrained language model, and for each layer $l$ of model $m$, we build TR-level embeddings. Finally, to account for the hemodynamic delay, we concatenate the previous four TR embeddings to form the final input $\mathbf{X}_l^m$ of shape $(K, 4H)$, with $K$ and $H$ being the number of time points (TRs) and the hidden dimension of layer $l$, respectively. We provide more details on how to obtain TR-level embeddings, together with illustrations, in Appendix A.4.

We then pass the inputs $\mathbf{X}_l^m$ into a trained brain encoding model, which is a ridge-regularized linear regressor $f : \mathbf{X}_l^m \to y_i^j$ that maps the LLM-derived input $\mathbf{X}_l^m$ to the voxel[1] activity $y_i^j$. Following prior work, we use 4-fold cross-validation for ~~HP~~Harry Potter, and 11-fold cross-validation (with each fold corresponding to one story) for ~~MRH~~Moth Radio Hour, and select the regularization strength via nested cross-validation (Jain & Huth, 2018b; Toneva & Wehbe, 2019; Schrimpf et al., 2021; Oota et al., 2023a).

**Attribution approach.** To make the encoding weights usable in an attribution setting, we decompose the learned weight matrix of shape $(4H, V_i)$, with $V_i$ being the number of voxels recorded for subject $i$, into four matrices of shape $(H, V_i)$, one per TR. Each of these matrices is used to initialize a linear projection layer $g$ that maps the LLM representation at a given TR to voxel activity.

For each TR, we extract the input contexts corresponding to the words shown in that time window. We process each context with the LLM and average the token embeddings of the final word to obtain word-level embeddings. These are then averaged to form a TR-level embedding of shape $(H, )$. The resulting representation is passed through the linear projection layer $g$ to produce predicted voxel activity $\hat{y}_i^j$ across all voxels. We compute the mean squared error ~~(MSE)~~ between the predicted and true voxel activity:

$$MSE = \frac{1}{V_i} \sum_{j=1}^{V_i} (y_i^j - \hat{y}_i^j)^2$$

We then apply the attribution method with respect to this loss to compute token-level importance scores for each input word, as shown in Figure 1. To obtain word-level attributions, we sum the token-level scores corresponding to each word, as is standard practice in the literature (Dürlich et al., 2025). This aggregation is particularly appropriate for ~~IG~~Integrated Gradients, as it preserves the completeness property of the method (Sundararajan et al., 2017) (i.e., the sum of the attribution scores for an input sequence should be equal to the difference in model prediction for the real input sequence and the baseline). Since each input word may appear in multiple overlapping contexts across the 4 concatenated TRs, we sum all its associated attribution values to obtain a final word-level score per TR.

### 3.3.2 END-TO-END INPUT ATTRIBUTION FOR NEXT-WORD PREDICTION

To compare ~~input attributions for BA with those derived from NWP~~ BA and NWP attributions fairly, we ensure ~~both prediction tasks use equivalent~~ that both tasks use the same input information. For each TR, we ~~construct~~ build an extended context ~~composed of all words from the four input contexts of each of~~ that contains all words appearing in the four contexts associated with the four concatenated TRs used in the ~~BA task. We~~ brain encoding model.

---

[1] A voxel (volumetric pixel) represents a small 3D unit of brain tissue captured in an fMRI scan, recording a time series of blood-oxygen-level-dependent signals during the reading task.

For NWP, we use teacher forcing~~, a standard technique where, at each time step, the model is provided with the ground-truth input tokens to predict the next token. In our case, we use the extended input context (containing all words from the contexts of all the concatenated TRs)~~: the model receives the entire extended context as input and is asked to predict the ~~word that immediately follows it. Since most words are tokenized into~~ next word in the stimulus. Because many words ~~consist of~~ multiple sub-word ~~units~~tokens, we predict each token of the ~~target~~ next word and compute the ~~average~~ mean cross-entropy loss across ~~them. This scalar loss serves as the objective for attribution . This approach ensures that words composed of multiple tokens are treated fairly and that~~ those tokens. This single scalar loss is then used as the attribution ~~target. This ensures that~~ multi-token words are treated consistently and that the attribution reflects the ~~full predictive burden of the model. We then~~ model's full predictive load.

Finally, we compute token-level attributions using the same gradient-based methods ~~and sum them~~ as in BA and then sum token scores to obtain word-level attribution ~~scores~~values.

## 3.4 EVALUATION METRICS

To compare the attributions obtained for BA and NWP, we employ two complementary evaluation metrics: intersection over union (IoU) and center of mass (CoM). The IoU (Jaccard, 1908) measures the degree of overlap between the sets of words prioritized by the two tasks. Because attribution scores are continuous, we define "important words" by selecting the smallest subset of words whose cumulative attribution reaches a given threshold $t$. Let $\mathbb{W}_{brain}^t$ and $\mathbb{W}_{NWP}^t$ denote these sets of most important words for BA and NWP, respectively. The IoU is then defined as:

$$IoU^t = \frac{|\mathbb{W}_{brain}^t \cap \mathbb{W}_{NWP}^t|}{|\mathbb{W}_{brain}^t \cup \mathbb{W}_{NWP}^t|}$$

We compute IoU scores for $t = 1, 2, 3, 5, 10, 20, 30, 40, 50, 60, 70, 80, 90, 95, 98\%$ to assess how the overlap changes as larger subsets of words are considered.

The CoM, instead, captures where in the input context important words tend to occur (earlier vs. later positions). Formally, given a sequence of $n$ input words with corresponding attribution scores $a_1, \ldots, a_n$, the CoM is defined as:

$$CoM = \frac{\sum_{i=1}^{n} d_i \cdot a_i}{\sum_{i=1}^{n} a_i}$$

where $d_i$ denotes the distance from the most recent word in the input context, and $a_i$ is its corresponding attribution score. A lower CoM indicates that the task places more importance on words closer to the end of the input context, while a higher CoM reflects a focus on earlier words.

Because the CoM summarizes only the mean positional tendency of attributions, it cannot distinguish whether attribution mass is tightly concentrated around that mean or broadly distributed across the context. We therefore introduce an attribution-spread metric that captures the dispersion of important words over positions. Let $d_i$ denote the distance of word $i$ from the most recent token, and let $a_i \geq 0$ be its attribution score. After normalizing by total attribution mass, we compute the weighted positional standard deviation:

$$\sigma = \sqrt{\frac{\sum_{i=1}^{n} a_i(d_i - \text{CoM})^2}{\sum_{i=1}^{n} a_i}}$$

A low spread indicates that attribution is concentrated in a narrow region (e.g., a sharp recency peak), whereas a high spread indicates that attribution is distributed across a wider range of context positions (e.g., a broad recency effect or multiple peaks).

## 4 RESULTS

To investigate the relationship between BA and NWP, we perform attribution analyses for both prediction tasks to identify the most influential input words. We first analyze the overlap (IoU) between words that are important for NWP and BA, respectively. We then examine which words are

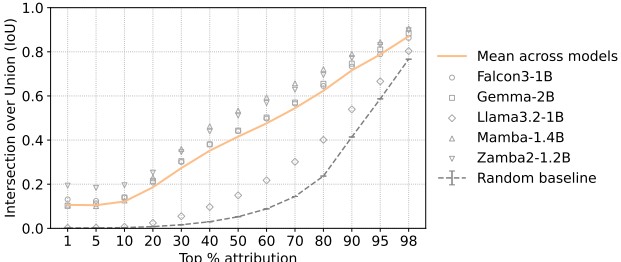

Figure 2: Intersection over Union (IoU) between BA and NWP ~~as a function~~ attributions considering top-attributed word sets of increasing size. For each context, the top-$t\%$ attribution ~~threshold for~~ set is defined as the ~~HP dataset~~smallest subset of words whose cumulative attribution mass reaches $t\%$ of the total. We report the average IoU across contexts, layers, subjects, and models, with gray shapes indicating per-model IoUs. The gray line represents a random baseline. The overlap between important words for the two tasks is very low for $t \leq 10\%$, and then grows linearly as the threshold is increased, but it is always above chance.

uniquely or jointly important across tasks, and their linguistic types (semantic, syntactic, discourse). Finally, we analyze positional patterns of high-attributed words, together with CoM values. To perform these analyses, we compute BA using LLM representations at every layer of each model, and then compute attributions at three representative layer depths (early, middle, late) by selecting the layer with highest BA at each depth. We provide more details on layer selection in Appendix B. As a sanity check to ensure that attribution scores highlight words important for the task at hand, we probe their functional relevance by masking top-attributed words. We find that even removing only the top $1\%$ of words virtually abolishes predictive power (see Appendix C). Having established that attribution scores capture meaningful signal, we present results at increasing levels of granularity.

**Low overlap between top-attributed words for BA and NWP.** After computing attributions for BA and NWP, for each context and model, we rank all words by their attribution scores and select the smallest subset that covers a cumulative attribution threshold of $t\%$. We then compute the overlap (i.e., IoU) between the top-attributed sets for BA and NWP, for different thresholds $t\%$. Figure 2 shows the results for all individual models, as well as their average, on the ~~HP~~ Harry Potter dataset. As a random baseline, for each TR and threshold $t$, we drew 100 pairs of random word sets matching the sizes of the BA-and NWP-top-$t\%$ sets, and averaged their IoUs. The baseline confirms that chance overlap is essentially zero at stringent thresholds ($t < 10\%$); our observed IoU of $\approx 0.16$ at this range therefore cannot be explained by random coincidence. Even as $t$ grows, empirical IoUs remain consistently 1.5–2$\times$ above chance (up to $t = 80\%$), indicating systematic, though limited, overlap between tasks. At low thresholds ($t \leq 10\%$), the overlap is minimal (IoU $\approx 0.1-0.2$), suggesting that the two tasks rely on distinct subsets of important words. As $t$ increases, IoU gradually rises, eventually exceeding $0.8$ at $t = 98\%$. This pattern indicates that while both tasks ultimately draw on broad context, their top attribution targets diverge significantly. All the observations we made hold on the ~~MRH~~ Moth Radio Hour dataset (see Appendix D.1).

**BA and NWP have opposing trends of attribution spread.** To show differences in how attribution spreads across context words for BA and NWP, Figure 3 reports the number of unique words needed to cumulatively reach increasing attribution thresholds ~~$t$ on the HP~~ $t\%$ on the Harry Potter dataset, along with the corresponding area under the curve (AUC). The results reveal a task-dependent shift in attribution spread across layers. At early layers, NWP requires significantly more unique words than BA to reach a given ~~$t$~~ $t\%$, consistent with its reliance on highly local cues, such as collocational associations and lexical repetition (Oh & Schuler, 2023). These local, word-level cues are already captured at early model layers (Mischler et al., 2024). In contrast, at middle and late layers, BA exhibits higher attribution spread, suggesting that it integrates higher-order linguistic structures, such as semantic and discourse-level information, that emerge later in the model (Merlin & Toneva, 2024; Mischler et al., 2024). This shift is also reflected in the AUC trends: the AUC for BA increases steadily from early to late layers, while the AUC for NWP decreases, highlighting that BA draws on increasingly distributed contextual evidence as representations deepen. These

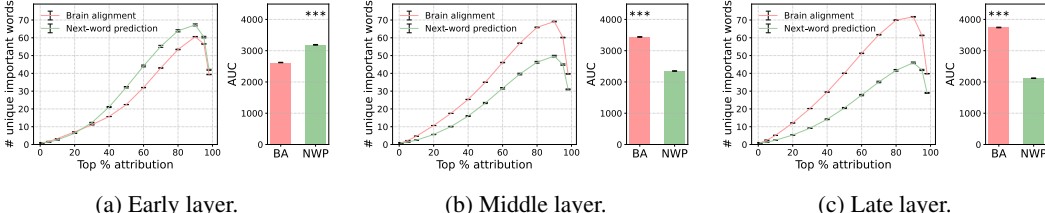

(a) Early layer.      (b) Middle layer.      (c) Late layer.

Figure 3: Number of top-attributed unique ~~important~~ words, averaged across models and contexts, that are needed to cumulatively reach ~~increasing~~ $t\%$ of the total context attribution~~thresholds~~, across three representative layers on the ~~HP~~ Harry Potter dataset. Error bars represent the standard error across contexts. Each plot compares BA and NWP, with the area under the curve (AUC) quantifying the total attribution spread. Asterisks denote significant differences ($p < 0.001$), assessed via a two-sided paired t-test, with Benjamini-Hochberg correction (Benjamini & Hochberg, 1995).

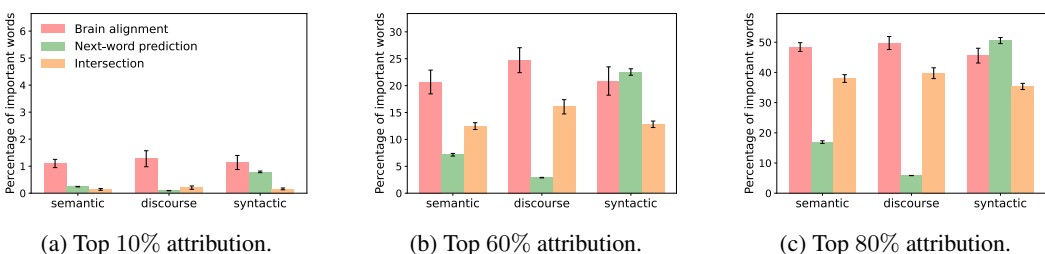

(a) Top $10\%$ attribution.      (b) Top $60\%$ attribution.      (c) Top $80\%$ attribution.

Figure 4: Distribution of top-attributed words across linguistic feature categories (semantic, discourse, syntactic) for BA-only, NWP-only, and for both. Results are averaged across layers, subjects, and models, with standard errors across models. We show results for the subsets of words that cumulatively account for the top $10\%$, $60\%$, and $80\%$ of each task's total attribution ~~scores~~mass. At lower thresholds, NWP prioritizes syntactic features, while BA emphasizes semantic and discourse-level content. As the threshold increases, the overlap between tasks grows, but BA consistently favors meaning-oriented features.

findings, which also hold for the ~~MRH~~ Moth Radio Hour dataset (see Appendix D.2), suggest that brain-aligned signals become more distributed and semantically rich at higher levels of processing, beyond what is required for surface-level prediction.

**NWP relies heavily on syntax, while BA also draws on semantics and discourse.** To better understand the types of linguistic information driving BA and NWP, we analyzed the distribution of top-attributed words across three story-level features in the ~~HP~~ Harry Potter dataset: semantic, discourse, and syntactic features. Figure 4 presents the percentage of words in each category that are uniquely or jointly important for BA and NWP, evaluated at three attribution thresholds ($t = 10\%, 60\%, 80\%$). For each context, the percentage for category $x$ is computed as the number of features in $x$ found in top-attributed words, divided by the total number of features in $x$. Words with multiple features contribute multiple counts, and computations are done independently per category. Figure 4 reports averages across models, layers, subjects, and contexts, with error bars denoting the standard error across models. A value of $100\%$ means that all features of category $x$ lie in top-attributed words on average. Across all thresholds, NWP shows a clear bias toward syntactic features, consistent with prior findings (Oh & Schuler, 2023). In contrast, BA exhibits a more balanced distribution: while syntactic cues remain important, as also shown by (Oota et al., 2023b), a substantial proportion of its attribution falls on semantic and discourse-level features. These findings suggest that fully trained LLMs prioritize different features for NWP and BA, in accordance with (AlKhamissi et al., 2025). While NWP relies more on syntax, it still captures semantic cues: at $t = 60\%$, it marks $\approx 20\%$ of feature words as important ($\approx 7\%$ uniquely, $\approx 13\%$ jointly), compared to $\approx 33\%$ for BA. Although lower, this is still substantial given that BA spreads importance across more words. Appendix E.3 shows the results hold with ~~IG~~Integrated Gradients, confirming they are not an artifact of the attribution method.

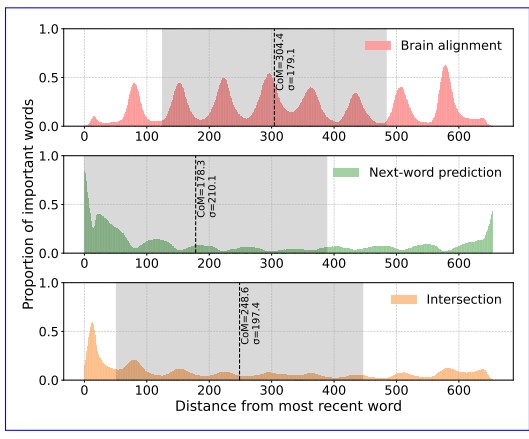 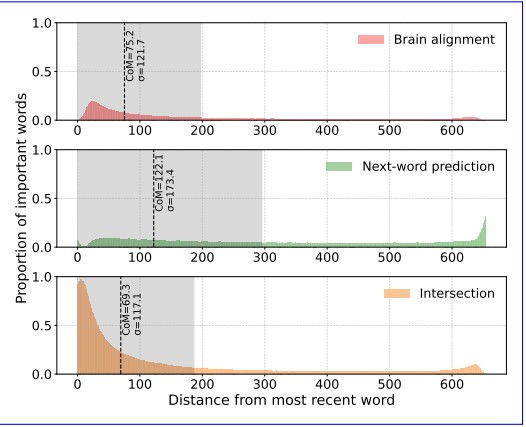

(a) Llama3.2-1B.  (b) Mamba-1.4B.

Figure 5: Distribution of ~~the~~ top-60% ~~attributed~~ most influential words, defined as the smallest subset whose cumulative attribution reaches 60% of the total, by ~~their~~ distance from the most recent word in the context ~~for two representative models~~(0 = last word). ~~For each model~~Each bin shows, ~~we plot~~ for that distance, the proportion of important words ~~located~~ relative to the total number of words occurring at ~~each distance bin, comparing BA and NWP~~that position. NWP ~~attributions peak at the beginning and end of contexts (~~exhibits strong primacy and recency ~~)~~peaks, whereas BA ~~peaks are~~ shows a broader~~and~~, less edge-focused recency profile. ~~The dashed~~ Dashed lines ~~represent~~ mark the CoM~~values~~, and shaded areas indicate weighted positional standard deviation, which ~~are shifted toward higher distances~~ is generally larger for ~~Llama3.2-1B~~NWP due to its sharp primacy peak.

**Fine-grained attribution reveals positional biases and model-specific integration strategies.** Our word-level attribution enables a detailed analysis of how models prioritize contextual information across different positions in the input sequence. Figure 5 reports the distribution of words included in the top-60% attribution set as a function of their distance from the most recent word (0 = ~~latest~~last word in the context, 736 = ~~farthest~~first word in the context, with 736 corresponding to the maximum extended context length after concatenating the 4 TRs to account for the BOLD delay). Dotted lines represent the CoM values~~.~~, while shaded regions show the weighted positional standard deviation. Note that the y-axis reflects the proportion of important words at each relative position (number of important words / total number of words), not the absolute attribution mass. Because there are shorter contexts at the beginning of each fold, distant positions occur in fewer examples. Consequently, even if peaks appear at large distances, they contribute far less total attribution mass than the dense cluster of important words near the most recent positions, keeping CoM values close to the recency end. However, the presence of a sharp primacy bias (i.e. peak at large distances) is reflected in a high weighted positional standard deviation. We show results for Llama3.2-1B and Mamba-1.4B, and report full model-wise and per-subject results on the ~~HP~~ Harry Potter dataset in Appendix F. Llama3.2-1B is the most brain-aligned model among the tested ones (see Appendix G) and exhibits a unique attribution pattern, while Mamba-1.4B reflects the typical trends seen in other models (Gemma-2B, Falcon3-1B, Zamba2-1.2B).

Across all models, NWP consistently shows a sharp bimodal distribution: a pronounced peak at the end of the context (low distance from most recent word) and a secondary peak at the beginning of the context (high distance from most recent word). These findings confirm the presence of recency and primacy biases in transformer-based models (Oh & Schuler, 2023; Liu et al., 2024) and SSMs (Wang et al., 2025; Morita, 2025), which might be due to: (1) architectural elements, such as positional encodings and attention, may favor the edges of the context (Su et al., 2024; Wu et al., 2025), or (2) biases in the training data, shaped by human communication patterns (Itzhak et al., 2024; Raimondi & Gabbrielli, 2025). We provide the first word-level comparison of primacy across both NWP and BA: results show that primacy bias is stronger in NWP attributions than in BA attributions, suggesting that the brain can modulate reliance on early context more flexibly, likely guided by the specific contextual content. Interestingly, BA displays a more pronounced recency bias than NWP in 4 out of 5 models, with a broader peak at low distances. This stronger recency focus is reflected

in the CoM analysis as, for most models, BA's CoM is slightly closer to the most recent word than that of NWP. However, the exact magnitude of this shift varies with the attribution threshold and model architecture, indicating differences in how each system integrates context. For $t = 10\%$, CoM values for BA, NWP, and their intersection are much closer to the most recent word, indicating a dominance of the recency bias over the primacy bias for both tasks (see plots in Appendix F.1).

Llama3.2-1B deviates from these described trends by exhibiting an oscillatory attribution pattern, with pronounced peaks at non-contiguous distances. To ~~confirm that this unique pattern does not depend~~ assess whether this oscillatory structure depends on the attribution method, ~~in AppendixE.4 we provide results for Llama3.2-1B and Gemma-2B obtained using IG, which replicate~~ Appendix E.4 presents results obtained using Integrated Gradients and SmoothGrad. The former reproduces the same positional ~~patterns. However~~pattern, whereas the latter largely smooths out these oscillations. Moreover, additional experiments suggest that the oscillatory behavior of Llama3.2-1B is not an invariant property of the architecture. First, Qwen2-1.5B, which shares several key architectural features with Llama3.2-1B (RoPE, GQA, FlashAttention2, and quantization type), does not display oscillatory attributions (see Appendix H). Second, when replicating the positional pattern analysis on the ~~HP~~ Harry Potter dataset with a shorter context length (80 words) the oscillations for Llama3.2-1B disappear, yielding a smoother recency profile (see Appendix I). Finally, experiments on the ~~MRH~~ Moth Radio Hour dataset in Appendix D.3 also show no oscillatory behavior for Llama3.2-1B, instead producing the more conventional bimodal distribution. Taken together, these findings indicate that the oscillatory pattern ~~may be~~ reflects a stimulus- and context-dependent interaction amplified by raw gradient-based attributions, rather than a ~~direct consequence~~ stable architectural feature of Llama3.2-1B~~'s architecture.~~. When either the stimulus structure changes (MRH, shorter contexts) or local gradient noise is suppressed (SmoothGrad), the oscillations largely disappear. This suggests that the oscillatory peaks reflect how gradient-based methods react to regularities in the Harry Potter text. When these fine-grained gradient fluctuations are smoothed, the oscillations diminish, pointing to a stimulus-dependent rather than architecture-dependent origin.

## 5 DISCUSSION

Our work ~~introduces a tool for fine-grained analysis of BA and applies it to the study of~~ presents a pipeline that integrates established attribution methods into the brain-LLM alignment setting and uses it to study the relationship between BA and NWP. Using gradient-based word-level attributions, we show that the words important for predicting brain activity and the next word, respectively, differ substantially. This low overlap supports prior findings that brain-LLM alignment stems from deeper, semantically meaningful representations, rather than only predictive processing (Merlin & Toneva, 2024). We further demonstrate that BA has higher attribution spread compared to NWP at middle and late layers, while the opposite holds at early layers. This may due to the fact that NWP relies on highly localized signals, such as collocational associations and syntactic dependencies, that are better captured by early layers compared to higher-level semantic information (Mischler et al., 2024). We confirmed this hypothesis through our feature-based analysis, which shows that while both tasks attend to syntactic information, according to Oota et al. (2023b) and (Oh & Schuler, 2023), BA more strongly emphasizes semantic and discourse-level content. Finally, our positional attribution analysis highlights task-specific distributions: NWP exhibits recency and primacy biases across architectures, while BA shows a more pronounced and broad recency bias.

**Cross-architecture consistency and Llama3.2-1B's outlier behavior.** Across all models, Transformers (Llama3.2-1B, Gemma-2B, Falcon3-1B), state-space models (Mamba-1.4B), and hybrids (Zamba2-1.2B), we observe the same core attribution trends: (1) very low BA–NWP overlap, (2) wider NWP attribution spread in early layers and wider BA spread in middle and late layers, and (3) a robust bimodal NWP positional profile (primacy + recency) contrasted with a broader recency-focused BA distribution. The cross-architecture stability of these effects suggests that BA–NWP differences on the two analyzed datasets reflect task-driven functional usage of shared representations rather than architectural idiosyncrasies.

Llama3.2-1B is the only model exhibiting an oscillatory attribution pattern on the Harry Potter dataset. As shown above, this effect is anti-phased across tasks and reduces BA–NWP overlap without reflecting weaker brain alignment (indeed, Llama3.2-1B is the most brain-aligned model; Appendix G). Follow-up analyses (Appendix E.4) show that the oscillations persist under Integrated

Gradients, but disappear with shorter context windows and on the Moth Radio Hour dataset, and are not present in Qwen2-1.5B despite substantial architectural similarity. These findings indicate that the oscillatory behavior arises from an interaction between Llama's long-range positional encoding and the structural regularities of the Harry Potter text, rather than from architecture alone. We highlight this as an interesting stimulus–model interaction for future work.

**Limitations.** One limitation of our approach lies in the reliance on gradient-based attribution methods, which may be sensitive to local nonlinearities and model smoothness. We mitigate this issue by validating consistency across models with diverse architectures, and applying multiple attribution methods to confirm outlier patterns. ~~Another~~ A second limitation is that~~the~~, although the pipeline itself is model- and dataset-agnostic, our empirical evaluation covers only two narrative fMRI datasets, and therefore we refrain from making broad claims about generality across modalities or linguistic domains. Additionally, discourse feature annotations used in our analysis on the ~~HP~~ Harry Potter dataset are relatively coarse and limited to predefined categories. Still, they provide a useful first approximation for characterizing broad functional differences between tasks. Lastly, we perform attributions using frozen models, without optimizing them for BA. Consequently, our findings reflect the models' inductive biases, rather than an optimal solution, but this allows us to interpret how these systems process language out of the box.

**Future work.** Future work should apply our framework to larger and more diverse models (e.g., trained with instruction-following objectives), or models that have been specifically fine-tuned for BA (i.e., brain-tuned (Moussa et al., 2025)), enabling direct comparison between native and optimized representations. Additionally, if applied to alignment trajectories during training, our pipeline would allow to analyze the evolution of the relationship between NWP and BA (AlKhamissi et al., 2025). Our framework is also directly applicable to broader datasets and modeling paradigms, and we view expanding this analysis to non-narrative and multilingual datasets as an important next step. Furthermore, while we used our feature-based analysis to capture the major functional distinctions between BA and NWP, it would be interesting to extend it to more fine-grained semantic subcategories. Finally, linking attribution dynamics to behavioral or cognitive measures, such as comprehension or memory recall scores, may provide a richer understanding of how LLMs approximate human-like language processing.

## 6 CONCLUSION

~~We introduce the first end-to-end attribution framework for~~ We apply attribution methods to brain-LLM alignment and analyze the resulting attribution patterns, enabling fine-grained analysis of the reasons for this alignment. As a case study, we use our method to investigate a contentious question in the literature: the relationship between BA and NWP. Our results show that while NWP provides a strong basis for alignment, it does not fully capture the linguistic signals relevant to predict brain activity. BA depends on a broader set of contextual cues and places greater weight on semantic and discourse-level information, reflecting more distributed and integrative processing. By uncovering differences in attribution spread, positional biases, and linguistic feature reliance across architectures, our approach also provides insights into how BA may emerge. These insights open new directions for interpreting brain-aligned LLMs and refining their objectives to better match human cognition. Ultimately, our work demonstrates the value of attribution in bridging artificial and biological language representations.

## ETHICS STATEMENT

While our method provides attribution maps indicating which input words contribute most to a model's prediction of a participant's brain activity, these maps do not enable reconstruction of private thoughts or preferences. The attribution patterns reflect sensitivity of a linear encoding model trained on group-level fMRI data, not subject-specific semantic decoding. Importantly, the fMRI datasets used in this study involve publicly released, anonymized data. Nonetheless, we acknowledge that increasingly fine-grained BA attributions raise questions about potential misuse, such as over-interpreting subject-level patterns as markers of personal thoughts or preferences. In our work, subject-level visualizations are strictly limited to positional attribution distributions—that is, how strongly each task relies on words at different distances in the context. These plots do

not reveal semantic content, personal preferences, or subject-specific vocabulary associations. All other analyses show aggregate linguistic and positional statistics, in order to avoid publishing any subject-identifiable semantic maps. Future applications of such techniques should incorporate privacy-preserving data handling, explicit consent, and appropriate governance around brain-data usage.

## REPRODUCIBILITY STATEMENT

In this paper, we fully describe our attribution framework for brain–LLM alignment and how to evaluate it. (1) Section 3 details the datasets, preprocessing steps, model families, encoding models, and all hyper-parameters required to reproduce brain alignment (BA) and next-word prediction (NWP) attribution analyses. (2) Sections 4 and 5, together with Appendices A, B, and C, provide complete descriptions of the evaluation pipelines, masking experiments, and metrics used in our study, as well as additional analyses on alternative datasets and attribution methods. (3) Appendix J reports all details on compute time and resources. (4) We will release open-source code, including data preprocessing scripts, attribution implementations, and evaluation procedures, upon acceptance of the paper, available at (see supplemental - GitHub link available after double-blind review).

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

## A METHODS

### A.1 FMRI DATASETS

**Moth Radio Hour dataset.** The Moth Radio Hour ~~(MRH)~~ dataset contains naturalistic language stimuli in the form of 10 autobiographical stories (10–15 minutes each) from *The Moth Radio Hour* podcast. An additional 10-minute story, presented twice to each participant, serves as a validation set. Brain responses were recorded from 9 subjects while they either listened to or read the stories. In this work, we analyze only the reading condition. Adding listening would introduce a second

modality and additional variability that falls beyond our current scope. The words of each story were presented one-by-one for a duration that is equal to the duration of that word in the spoken story, and the functional image acquisition rate is $TR = 2.0045s$. Brain responses were collected in two 3-hour scanning sessions, performed on different days.

**Harry Potter's word-level annotations.** Each word in the ~~HP~~ Harry Potter dataset is annotated as one or more high-level linguistic features belonging to 3 categories: semantic, syntactic, and discourse-level features. Semantic annotations consist of a 100-dimensional vector per-word representing different word meanings, computed for each word in the chapter using the co-occurrence patterns with other words in a large text corpus. Syntactic labels consist of 45-dimensional vectors representing 28 POS and 17 dependency relationships, obtained using an automated parser, while discourse-level annotations consist of a 27-dimensional vector representing ~~the following features~~ several broad semantic categories that identify story-specific, high-level content, including: verbs (e.g., 'be', 'hear', 'know'), speech ('speak'), motion ('fly', 'manipulate', 'move'), emotion (e.g., 'annoyed', 'dislike', 'fear', 'like'), and characters (e.g., 'draco', 'filch', 'harry', 'ron'). Each word in the chapter may be associated with zero, one, or multiple features within each category (semantic/syntactic/discourse).

## A.2 MODELS DESCRIPTION

In our experiments, we compare results obtained using five different models, belonging to three model families: transformers, SSMs, and hybrid. We herein provide a detailed description of each of the used models:

- Falcon3-1B Team (2024) is a decoder-only transformer with 18 layers and 2048-dimensional representations. It uses Grouped-Query Attention (GQA) (8Q/4KV heads of size 256) and rotary positional encodings (RoPE) Su et al. (2024) with a very large $\theta = 1000042$, enabling sequences up to $4K$ tokens. The 1B model was distilled and depth-pruned from a 3B teacher and trained on $\approx 80G$ tokens of multilingual web, code and STEM text, with SwiGLU activations and RMSNorm layers.

- Gemma-2B Mesnard et al. (2024) is an 18-layer, 2048-hidden-size decoder-only transformer. It employs multi-query attention (MQA) (8Q/1KV heads of size 256) and standard RoPE ($\theta$=10000) for contexts of up to $8K$ tokens. The 2B checkpoint is distilled from a 9B teacher and pretrained on a $\approx 3T$-token filtered web+code mix. It leverages GeLU activations and RMSNorm layers.

- Llama3.2-1B Meta AI (2024) consists of 16 transformer blocks with hidden size 2048. It uses GQA (32Q/8KV heads), RoPE scaled to handle $128K$-token windows, and SwiGLU activations. Pretraining covered up to $9T$ tokens of multilingual/coding text, with logits-distillation from Llama3.1-8B and -70B.

- Mamba-1.4B Gu & Dao (2023) is a pure selective state-space model, using no self-attention. It stacks 48 SSM blocks with embedding size 2048, giving linear training cost and constant-time generation. Benchmarks show accurate extrapolation beyond $1M$ tokens. The public 1.4B checkpoint was trained on $\approx 1T$ tokens of English/Russian web data.

- Zamba2-1.2B Glorioso et al. (2024) is a 38-layer hybrid model combining SSM and attention: a Mamba2 Gu & Dao (2023) backbone interleaves with six weight-shared attention blocks. Similarly to all the other models we consider, the hidden size is 2048. Attention layers use RoPE and low-rank adapters (LoRA) Hu et al. (2022) on both the shared attention and shared MLPs for per-depth specialization. The model supports $4K$-token contexts and was trained on $3T$ tokens of Zyda-2 web text and code, then annealed on $100B$ curated tokens.

## A.3 GRADIENT-BASED ATTRIBUTION METHODS

We employ two widely used gradient-based attribution methods to compute input importance scores: Gradient × Input and Integrated Gradients. Both methods quantify the contribution of each input token to a model's output by analyzing how sensitive the output is to changes in the input. These methods are implemented using the Captum library Kokhlikyan et al. (2020).

**Gradient × Input** ~~(GXI)~~.   Gradient × Input Shrikumar et al. (2016) is a simple yet effective attribution method. For a given input $\mathbf{x} = (x_1, \ldots, x_n)$ and model output $F(\mathbf{x})$ (e.g., a scalar loss), ~~GXI~~ Gradient × Input assigns importance to input component $x_i$ as:

$$GXI(x_i) = \frac{\partial F(\mathbf{x})}{\partial x_i} \cdot x_i$$

The attribution score is the element-wise product of the input embedding and the gradient of the output with respect to that input. This method captures both the sensitivity of the model's output to each input dimension and the magnitude of the input itself. ~~GXI~~ Gradient × Input is computationally efficient and has been shown to produce biologically meaningful attributions in recent brain alignment work Rahimi et al. (2025).

**Integrated Gradients** ~~(IG)~~.   Integrated Gradients Sundararajan et al. (2017) address the limitations of standard gradients by accumulating gradients along a straight-line path from a baseline input $\mathbf{x}'$ to the actual input $\mathbf{x}$. The attribution score for input component $x_i$ is defined as:

$$IG(x_i) = (x_i - x_i') \cdot \int_{\alpha=0}^{1} \frac{\partial F(\mathbf{x}' + \alpha(\mathbf{x} - \mathbf{x}'))}{\partial x_i} \, d\alpha$$

In practice, the integral is approximated using a Riemann sum with $m$ interpolation steps:

$$IG(x_i) \approx (x_i - x_i') \cdot \frac{1}{m} \sum_{k=1}^{m} \frac{\partial F(\mathbf{x}' + \frac{k}{m}(\mathbf{x} - \mathbf{x}'))}{\partial x_i}$$

We use the zero embedding as the baseline $\mathbf{x}'$ and set $m = 20$ in our experiments. ~~IG~~ Integrated Gradients is more computationally expensive than ~~GXI~~ Gradient × Input due to multiple forward and backward passes, but it provides a more theoretically grounded attribution estimate.

~~We primarily use GXI due to its efficiency, and validate select findings using IG~~

**SmoothGrad.**   SmoothGrad (Smilkov et al., 2017) is a noise-averaging technique designed to reduce the high-frequency noise often present in gradient-based attribution maps. Rather than modifying the attribution method itself, SmoothGrad operates by repeatedly adding Gaussian noise to the input and re-running a base attribution algorithm. In our experiments, we use Integrated Gradients as base technique. Formally, given an attribution method $A(\mathbf{x})$, SmoothGrad constructs noisy inputs $\mathbf{x} + \boldsymbol{\epsilon}_k$ with ~~Llama3.2-1B, we observe that IG produces qualitatively similar attribution patterns to GXI, supporting the robustness of our conclusions.~~ $\boldsymbol{\epsilon}_k \sim \mathcal{N}(0, \sigma^2 I)$, computes attributions for each noisy input, and averages them:

$$SG(x_i) = \frac{1}{K} \sum_{k=1}^{K} A(\mathbf{x} + \boldsymbol{\epsilon}_k)_i$$

where $K$ is the number of noise samples. We use Captum's implementation of SmoothGrad, which applies Gaussian noise to the inputs and computes smoothed attributions by averaging the resulting Integrated Gradients across the noisy samples. To ensure computational feasibility, we set $K = 5$ and the number of interpolation steps $m = 5$. SmoothGrad preserves the interpretability of the underlying attribution method while improving stability by averaging out local nonlinearities and gradient saturation effects.

### A.4   BRAIN ACTIVITY PREDICTION

**LLM representations.**   We begin by extracting contextual embeddings from a pretrained LLM (see Figure 6). For each word $w$ in the text, we construct a context of $L = 640$ words with $w$ as the final word. These contexts are fed into a pretrained language model, and for each layer $l$ of model $m$, we extract token embeddings of shape $(N, T, H)$, where $N$ is the number of input words, $T$ the

number of tokens in each context, and $H$ the hidden dimension. To obtain word-level embeddings, we average the token embeddings corresponding to the final word in each context, yielding a representation of shape $(N, H)$. Subsequently, we downsample the word-level embeddings to match the fMRI sampling rate by averaging the embeddings of all the words in each TR, obtaining TR-level embeddings of shape $(K, H)$. Finally, to account for the hemodynamic delay, we concatenate the previous four TR embeddings to form the final input $\mathbf{X}_l^m$ of shape $(K, 4H)$.

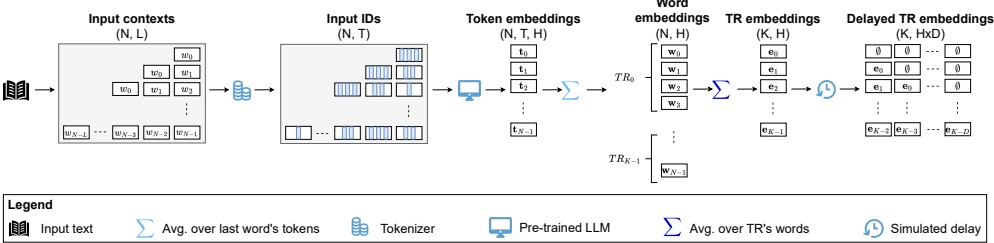

Figure 6: Extraction and processing of LLM representations. We associate a context of $L$ words to each word $w_i$ in the input text, with $i \in [0, N-1]$. We then pass the tokenized contexts to an LLM and extract token embeddings of shape $(N, T, H)$, with $T$ being the maximum number of tokens in a context. To get word embeddings of shape $(N, H)$, we average the embeddings of the tokens in the last word in each context. Finally, we average the word embeddings of all the words in each TR, obtaining TR embeddings of shape $(K, H)$, and concatenate $D$ previous TR embeddings to account for the delay in the BOLD response, yielding a final representation of shape $(K, H \times D)$.

**Brain encoding model.** We then pass the inputs $\mathbf{X}_l^m$ into a trained brain encoding model, which maps contextual LLM features to voxel-level brain activity. A voxel (volumetric pixel) represents a small 3D unit of brain tissue captured in an fMRI scan, recording a time series of blood-oxygen-level-dependent signals during the reading task. For each subject, we represent brain activity as a matrix $\mathbf{Y}_i$ of shape $(K, V_i)$, where $K$ is the number of time points (TRs) and $V_i$ is the number of voxels recorded for subject $i$. Each encoding model is a ridge-regularized linear regressor $f : \mathbf{X}_l^m \rightarrow y_i^j$ that maps the LLM-derived input $\mathbf{X}_l^m$ to the voxel activity $y_i^j$. Following prior work, we use 4-fold cross-validation for ~~HP~~Harry Potter, and 11-fold cross-validation (with each fold corresponding to one story) for ~~MRH~~Moth Radio Hour, and select the regularization strength via nested cross-validation, following prior work (Jain & Huth, 2018b; Toneva & Wehbe, 2019; Schrimpf et al., 2021; Oota et al., 2023a). The full brain alignment pipeline is illustrated in Figure 7.

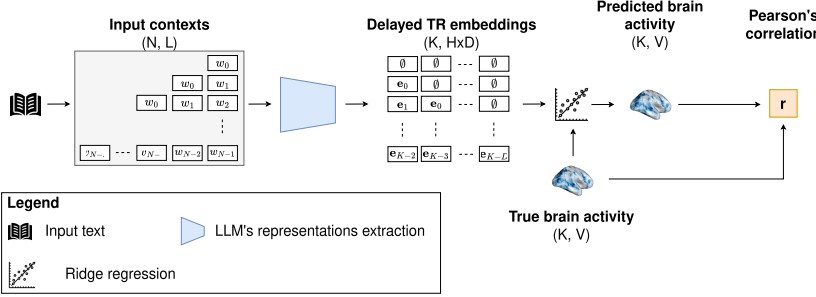

Figure 7: Illustration of the full brain alignment pipeline.

# B LAYER SELECTION FOR ATTRIBUTION

To investigate how attribution behavior varies across network depth, we compute BA attributions at three representative layers per model: one from early, one from middle, and one from late in the architecture. Rather than selecting fixed indices, we identify the most informative layers for BA. Specifically, for each model, we evaluate brain encoding performance at every layer using Pearson

Table 2: Indexes of the layers used for computing brain alignment attributions. For each model, $i \in [0, l - 1]$, with $l$ being the number of layers.

| Model | Layer depth | | |
|---|---|---|---|
| | Early | Middle | Late |
| Falcon3-1B | 5 | 9 | 12 |
| Gemma-2B | 5 | 6 | 12 |
| Llama3.2-1B | 4 | 9 | 12 |
| Mamba-1.4B | 15 | 29 | 32 |
| Zamba2-1.2B | 11 | 17 | 30 |

correlation between predicted and recorded fMRI activity. We divide each model into three equal-depth sections and select the layer within each section that achieves the highest average correlation across voxels and subjects. Table 2 lists the selected layers for each model.

## C    FUNCTIONAL IMPACT OF MASKING TOP-ATTRIBUTED WORDS

To ~~evaluate~~ assess whether words with high attribution scores are functionally important for each task, we ~~performed targeted~~ conducted masking experiments on the ~~HP dataset . Specifically, we replaced the~~ Harry Potter dataset under three complementary perturbation modalities:

- **Top-attributed → Random**: substitute top-attributed words with random words from the same chapter. This modality preserves lexical frequency and topic statistics, but might disrupt syntax and semantics.
- **Top-attributed → Same POS**: substitute top-attributed words with random words from the same chapter ~~and measured the resulting drop in predictive performance.~~ having the same POS tag, aiming at preserving syntax. This perturbation might still disrupt semantics.
- **Random → Same POS** (baseline): substitute randomly selected words with random words from the same chapter having the same POS tag.

For NWP, we report changes in cross-entropy (CE) loss relative to the baseline model performance. For BA, we quantify the decrease in Pearson's $r$ across language-selective ROIs, averaged across models and subjects. Together, these analyses confirm that top-attributed words indeed carry essential information for both tasks. In the following, we show preliminary results on a subset of the models: Gemma-2B and Falcon3-1B.

**Next-word prediction.** ~~Masking the most attributed words leads to sharp increases in CE across all models(Figure 8). Even masking only the top 1% of words more than doubles the loss, and increases remain above 100% for larger thresholds. This consistent degradation demonstrates that attribution scores capture words that are indispensable for predicting the next token, validating their functional relevance to the NWP objective~~ Figure 8 reports the mean percentage increase in cross-entropy loss across masking modalities, with error bars showing the standard error across models. Across all models and thresholds, masking top-attributed words with random words, either with different or same POS tag, leads to substantial increases in cross-entropy loss, confirming that the words highlighted by attribution are causally important for NWP. By contrast, masking the same number of randomly selected words causes only minor cross-entropy increases, confirming that the degradation observed in the top-attributed conditions reflects true causal importance, not the mere removal of any words.

**Brain alignment.** ~~For BA, performance collapses almost completely when only the top 1% of attributed words are masked (Figure 9 ). Across all ROIsand models, correlations with brain activity drop by nearly 100%, showing that the words flagged by attribution are critical for predicting neural responses. Increasing the threshold beyond 1% does not further reduce performance substantially, as alignment is already abolished by removing this minimal but highly informative subset of words~~

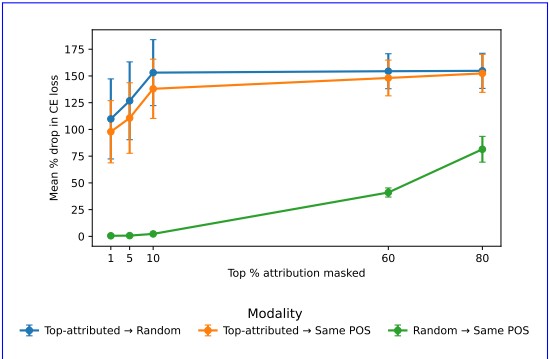

Figure 8: ~~Relative~~ Mean percentage increase in cross-entropy loss ~~when~~ across masking modalities. Results are averaged over models; error bars show ~~the top-$t\%$ most attributed words for each model~~ standard error across models. Masking ~~even 1% of~~ top-attributed words ~~leads to a substantial degradation in NWP performance~~, ~~highlighting the functional importance of top-attributed~~ either with random tokens or with same-POS replacements, results in large degradation, while masking random words has minimal effect.

Figure 9 reports the mean absolute drop in Pearson's correlation across ROIs, averaged across models. To quantify the significance of this drop, we report the original correlations computed for the unmasked inputs as dashed lines for each ROI.

or BA, all three modalities produce a substantial drop in alignment, but with a different pattern than NWP. The decrease in $r$ is large even under random masking, reflecting the fact that BA depends on coherent semantic context distributed across many words. Top-attributed masking (both random replacement and POS-controlled) generally yields equal or slightly greater degradation compared to the random baseline, indicating that attribution highlights words with above-baseline importance, but that BA's reliance is more broadly distributed and more semantically redundant than NWP's.

## D    RESULTS ON THE MOTH RADIO HOUR DATASET

### D.1    IOU ANALYSIS

To investigate whether BA and NWP rely on similar words, we compute the IoU between their top-attributed subsets across attribution thresholds. For the ~~HP~~ Harry Potter dataset (see Figure 2), we showed that for low thresholds ($t \leq 10\%$), the overlap is minimal (IoU $\approx 0.1-0.2$), indicating that the two tasks depend on largely distinct subsets of words. As the threshold increases, the IoU steadily rises, eventually exceeding $0.8$ at $t = 98\%$, consistent with both tasks drawing on broad context at larger scales. The same qualitative trend holds on the ~~MRH~~ Moth Radio Hour dataset (Figure 10): very low agreement at small thresholds, followed by a gradual rise toward convergence. This replication strengthens our conclusion that BA and NWP diverge most strongly on the small subset of words each considers essential, while converging only when most of the context is included.

### D.2    ATTRIBUTION SPREAD ANALYSIS

To examine whether the differences in attribution spread generalize beyond the ~~HP~~ Harry Potter dataset, we repeated the analysis on the ~~MRH~~ Moth Radio Hour dataset. As before, we measure the number of unique words required to cumulatively reach increasing attribution thresholds for BA and NWP, and conduct the analysis at three representative depths, selecting in each case the layer with the highest BA. Results shown in Figure 11 closely mirror those obtained on the ~~HP~~ Harry Potter dataset. At early layers, NWP requires substantially more unique words than BA to reach a given threshold, consistent with its reliance on highly local lexical cues. In contrast, at middle and late layers, BA exhibits a broader attribution spread, indicating stronger integration of distributed semantic and discourse-level information. The AUC trends replicate those on the ~~HP~~ Harry Potter dataset: BA spread increases steadily with depth, while NWP spread decreases. These convergent

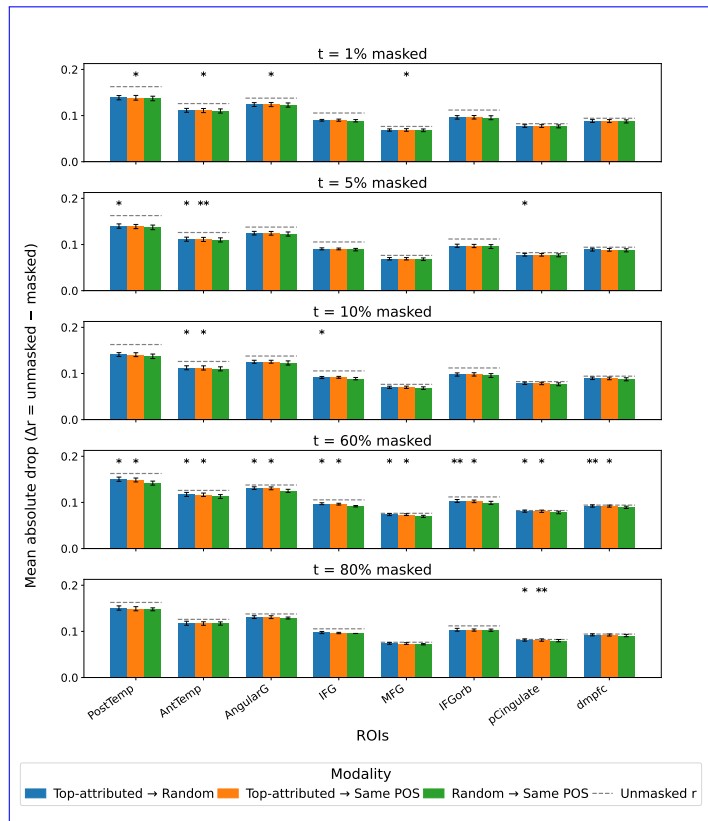

Figure 9: Mean ~~percentage~~ absolute drop in Pearson's $r$ across ~~language-selective ROIs after~~ masking ~~the top-$t\%$ most attributed words~~ modalities, grouped by ROI. ~~Results are averaged~~ Bars show averages across models, while dashed lines represent the original $r$ obtained using unmasked inputs. ~~Performance collapses almost completely after~~ Asterisks denote cases in which top-attributed random or POS-based masking ~~as little as 1% of~~ leads to significantly higher drops than the random masking baseline ($* : p < 0.05, * : p < 0.005, * : p < 0.001$). All perturbations strongly degrade alignment, but top-attributed ~~words~~ masking (random or same-POS) consistently matches or exceeds the random-masking baseline, demonstrating that ~~attribution scores identify~~ the top-ranked words ~~critical for BA~~ are indeed functionally informative.

results confirm that the opposing attribution spread dynamics of BA and NWP are robust across datasets and stimulus domains.

### D.3 POSITIONAL PATTERNS ANALYSIS

We repeated the positional attribution analysis on the ~~MRH~~ Moth Radio Hour dataset. Figures 12–14 report the distribution of top-attributed words at thresholds $t = 10\%, 60\%, 80\%$, plotted by distance from the most recent word in the context.

The results broadly replicate the patterns observed in the ~~HP~~ Harry Potter dataset. For both models, NWP shows a sharp recency bias, with a large fraction of attribution mass concentrated on the most recent words (i.e. lower distances). This is especially pronounced at low thresholds ($t = 10\%$), where almost all importance is assigned to words at the lowest distances. In contrast, BA exhibits a smoother and more distributed recency profile, with attribution spread over a broader set of nearby words. As the attribution threshold increases, BA continues to allocate importance across a wider temporal window, whereas NWP maintains its edge-focused profile.

Interestingly, Llama3.2-1B does not display the oscillatory attribution pattern for BA that we observed in the ~~HP~~ Harry Potter dataset. Instead, its BA attributions show a more conventional recency distribution, similar to Gemma-2B. This suggests that the oscillatory pattern may be stimulus-

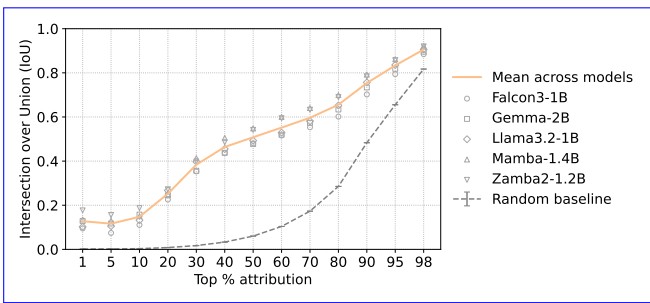

Figure 10: Intersection over Union (IoU) between BA and NWP ~~as a function~~ attributions considering top-attributed word sets of increasing size for the Moth Radio Hour dataset. For each context, the top-$t\%$ attribution ~~threshold for~~ (x axis) set is defined as the ~~MRH dataset~~ smallest subset of words whose cumulative attribution mass reaches $t\%$ of the total. We report the average IoU across contexts, layers, subjects, and models, with gray shapes indicating per-model IoUs. The gray line represents a random baseline. The overlap between important words for the two tasks is very low for $t \leq 10\%$, and then grows linearly as the threshold is increased.

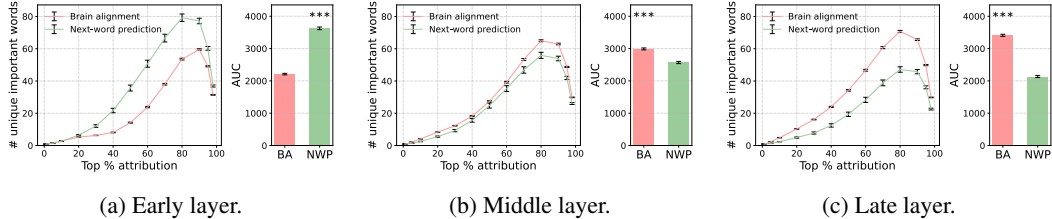

(a) Early layer.         (b) Middle layer.         (c) Late layer.

Figure 11: Number of unique important words, averaged across models and contexts, that are needed to cumulatively reach increasing attribution thresholds across three representative layers on the ~~MRH~~ Moth Radio Hour dataset. Error bars represent the standard error across contexts. Each plot compares brain alignment (BA) and next-word prediction (NWP). The area under the curve (AUC) quantifies the total attribution spread. Asterisks denote significant differences ($p < 0.001$), assessed via a two-sided paired t-test, with Benjamini-Hochberg correction (Benjamini & Hochberg, 1995).

dependent, potentially tied to the structural or lexical properties of the Harry Potter text rather than an invariant architectural feature of the model.

Overall, these findings confirm that the positional biases of NWP and BA are robust across datasets, while also revealing that certain model-specific patterns, such as Llama's oscillatory behavior, do not necessarily generalize across stimulus domains.

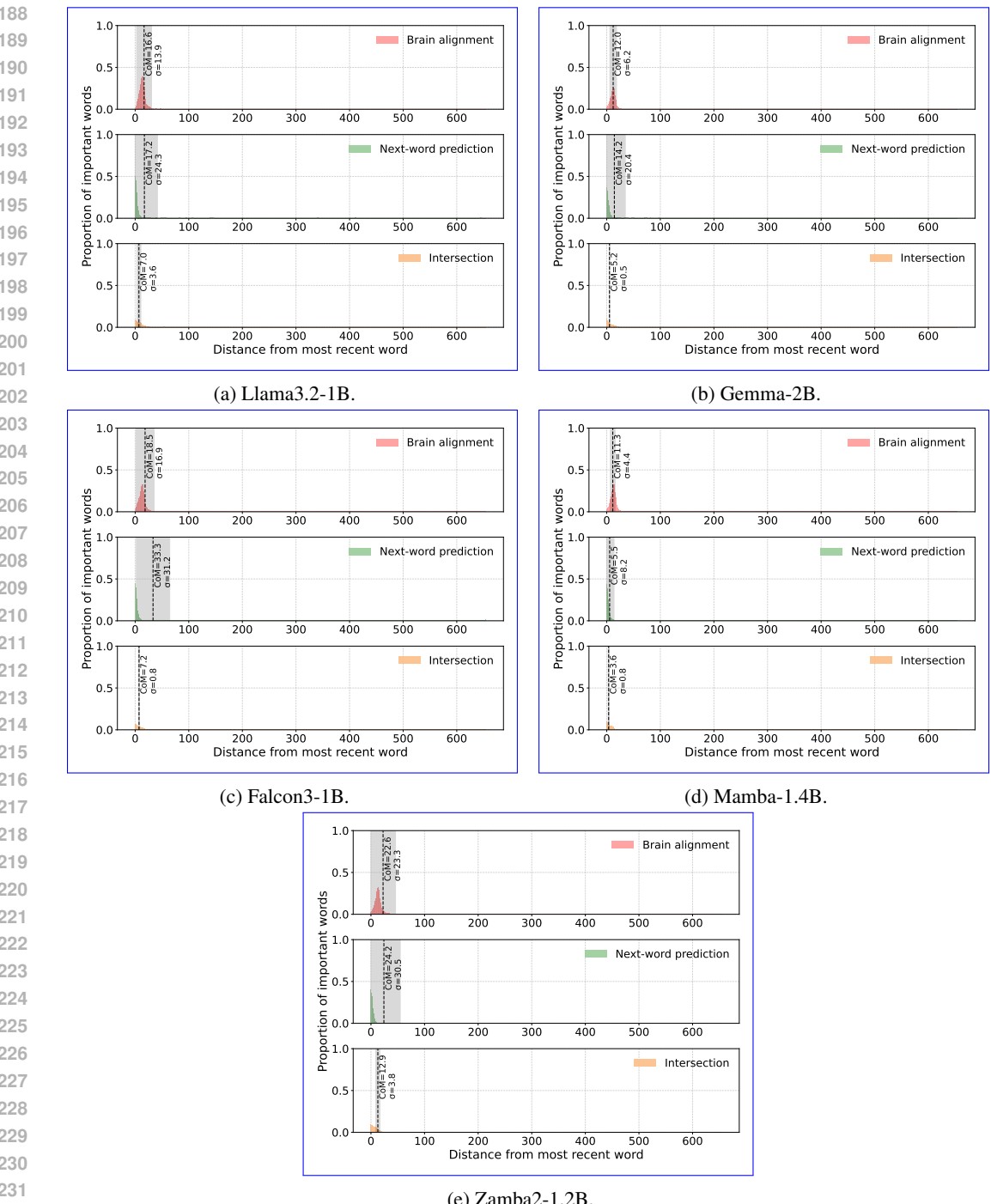

(a) Llama3.2-1B.

(b) Gemma-2B.

(c) Falcon3-1B.

(d) Mamba-1.4B.

(e) Zamba2-1.2B.

Figure 12: Distribution of top-attributed words (top 10% attribution) by distance from the most recent word. For each model, we plot the proportion of important words located at each distance bin, comparing BA and NWP. Both BA and NWP show a strong recency bias, with top-attributed words placed at very low distances from the most recent word.

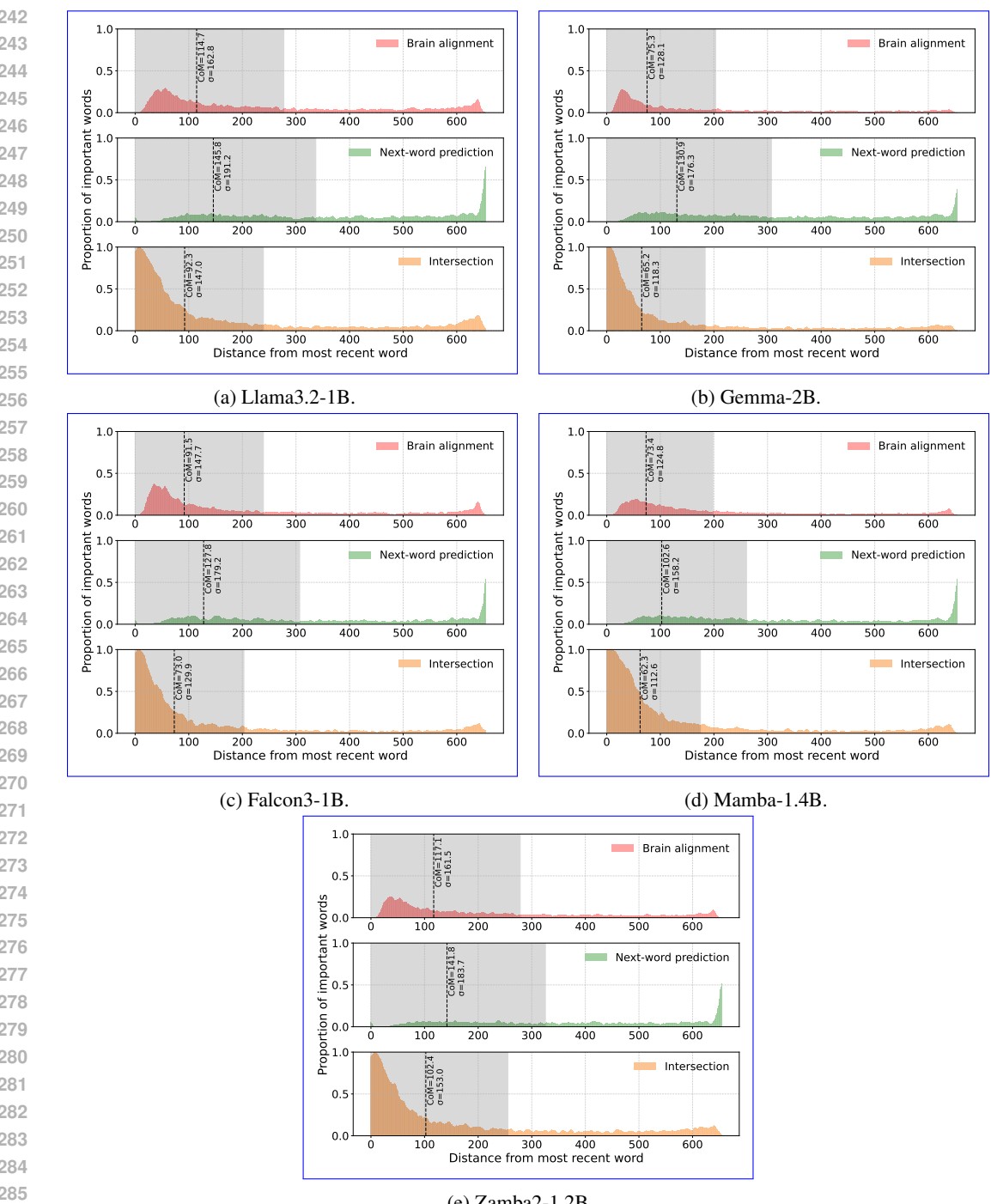

(a) Llama3.2-1B.

(b) Gemma-2B.

(c) Falcon3-1B.

(d) Mamba-1.4B.

(e) Zamba2-1.2B.

Figure 13: Distribution of top-attributed words (top 60% attribution) by distance from the most recent word. For each model, we plot the proportion of important words located at each distance bin, comparing BA and NWP. NWP shows a bimodal distribution, with sharp recency and primacy peaks. BA, on the other hand, emphasizes more distributed words, showing a much broader recency peak.

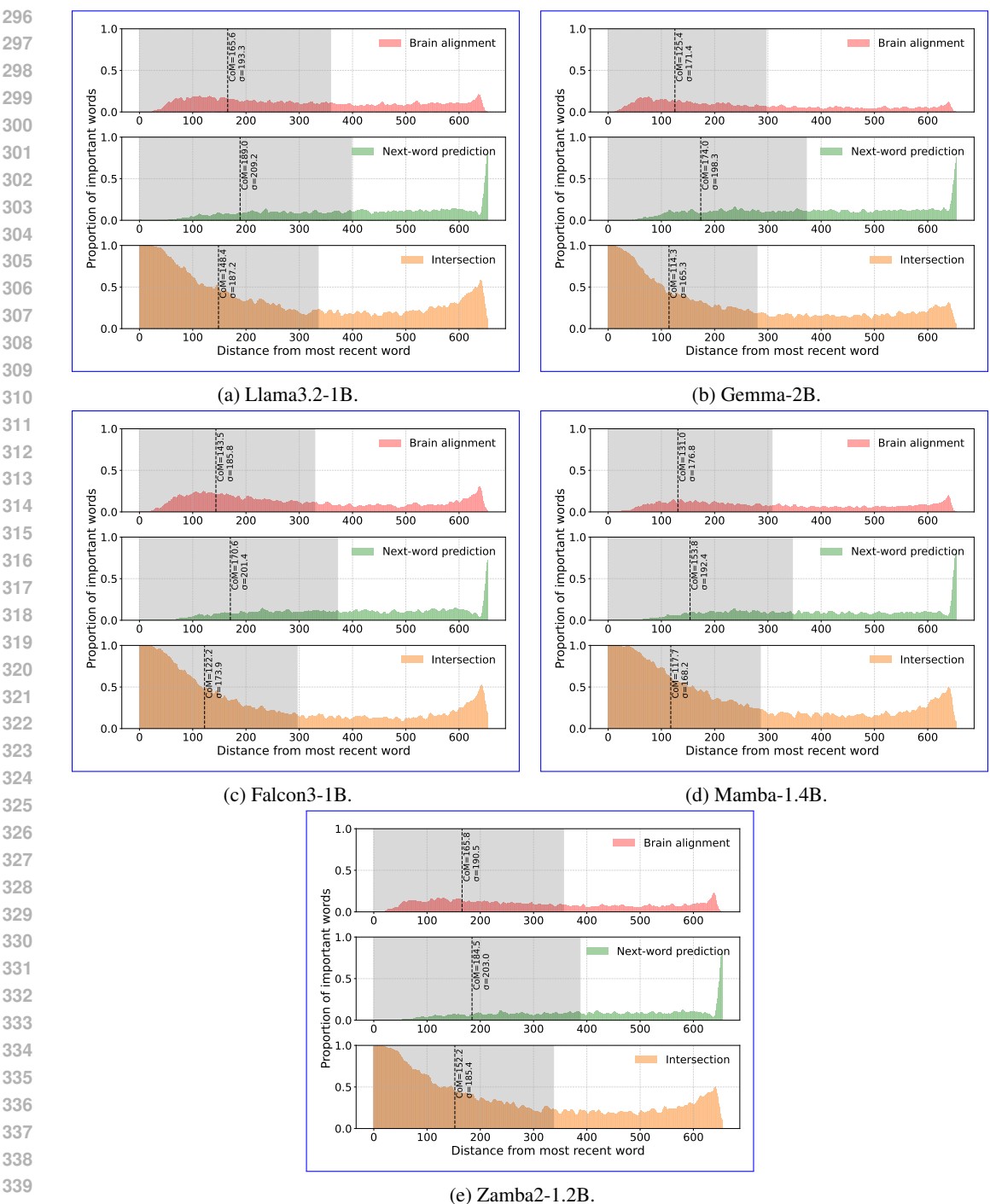

(a) Llama3.2-1B.

(b) Gemma-2B.

(c) Falcon3-1B.

(d) Mamba-1.4B.

(e) Zamba2-1.2B.

Figure 14: Distribution of top-attributed words (top 80% attribution) by distance from the most recent word. For each model, we plot the proportion of important words located at each distance bin, comparing BA and NWP. NWP shows a bimodal distribution, with sharp recency and primacy peaks. BA, on the other hand, emphasizes more distributed words, showing a much broader recency peak.

# E ROBUSTNESS TO ATTRIBUTION ~~ANALYSES USING IG~~ METHOD

To test the robustness of our results with respect to the attribution method, we also compute attributions on the ~~HP dataset using IG for~~ Harry Potter dataset using Integrated Gradients (Llama3.2-1B and Gemma-2B) and SmoothGrad (Llama3.2-1B, Gemma-2B, ~~the model displaying a unique oscillatory attribution distribution,~~ Falcon3-1B, Mamba-1.4B). In the following, we report full attribution analyses for both methods.

Consistent with our Gradient × Input results, these new results reveals that:

- BA and NWP rely on largely distinct sets of input words, with low IoU across thresholds.
- NWP attributions exhibit strong recency and primacy peaks, whereas BA integrates information more broadly across the context.
- The attribution-spread differences between BA and NWP persist, with NWP showing narrower distributions in early layers and BA displaying larger spread in middle-to-late layers.
- Model-specific patterns are preserved, including the oscillatory profile of Llama3.2-1B on the HP dataset and the absence of oscillations on MRH and on shorter context windows.

## E.1 IOU ANALYSIS

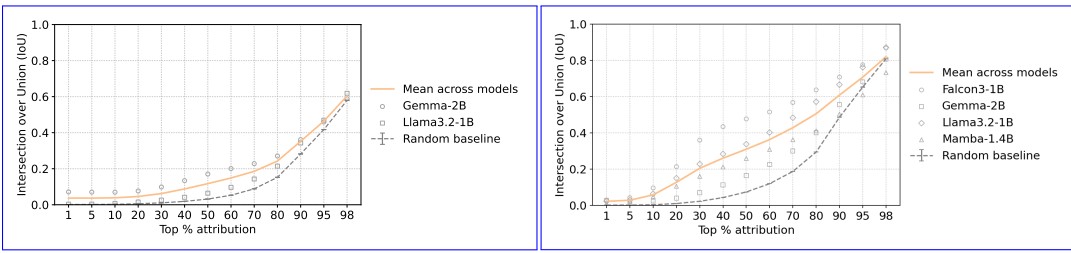

(a) Integrated Gradients.  (b) SmoothGrad.

Figure 15: Intersection over Union (IoU) between BA and NWP attributions considering top-attributed word sets of increasing size for the Harry Potter dataset using Integrated Gradients. For each context, the top-$t\%$ attribution (x axis) set is defined as the smallest subset of words whose cumulative attribution mass reaches $t\%$ of the total. We report the average IoU across contexts, layers, subjects, and models, with gray shapes indicating per-model IoUs. The gray line represents a random baseline. The overlap between important words for the two tasks is very low for $t \leq 10\%$, and then grows linearly as the threshold is increased, but it is always above chance.

To confirm that our attribution-level comparison between BA and NWP is not specific to Gradient × Input, we additionally compute IoU curves using Integrated Gradients and SmoothGrad attributions. Figure 15 shows the IoU between BA- and ~~Gemma-2B, which is representative of the behavior of all other models.~~ NWP-important words as a function of the top-t% attribution threshold, averaged across models, layers, subjects, and contexts.

The patterns for both attribution methods mirrors the Gradient × Input results: (1) IoU is extremely low for very small attribution budgets ($t < 10\%$), indicating that the highest-attribution words for BA and NWP are almost completely disjoint; (2) IoU increases smoothly as the attribution threshold grows, reflecting that broader sets of lower-attribution words are more likely to be shared.

## E.2 ATTRIBUTION SPREAD ANALYSIS

We next extend the attribution-spread analysis to Integrated Gradients and SmoothGrad. For each task, we compute the number of unique important words needed to cumulatively reach increasing attribution thresholds, providing a direct measure of how concentrated or distributed the attributions are. Larger curves indicate broader attribution spread.

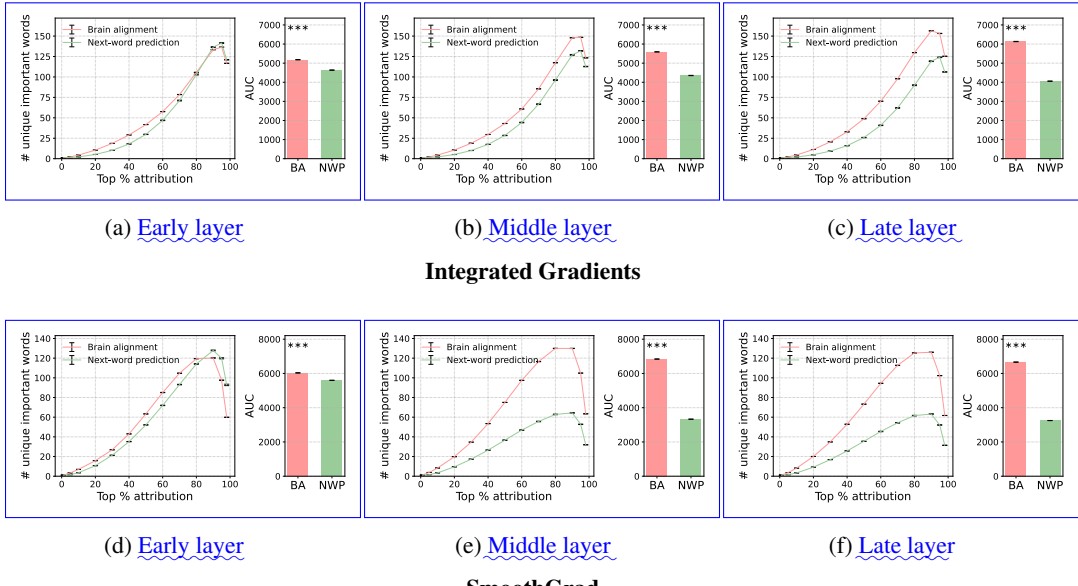

(a) Early layer  (b) Middle layer  (c) Late layer

**Integrated Gradients**

(d) Early layer  (e) Middle layer  (f) Late layer

**SmoothGrad**

Figure 16: Number of unique important words, averaged across models and contexts, that are needed to cumulatively reach increasing attribution thresholds across three representative layers on the Harry Potter dataset. Attributions are computed using either Integrated Gradients (IG) or SmoothGrad (SG). Error bars represent the standard error across contexts. Each plot compares BA and NWP, with the area under the curve (AUC) quantifying the total attribution spread. Asterisks denote significant differences ($p < 0.001$), assessed via a two-sided paired t-test, with Benjamini–Hochberg correction (Benjamini & Hochberg, 1995).

Figure 16 reports these curves for early, middle, and late layers, averaged models, for Integrated Gradients and SmoothGrad, respectively. We observe the same qualitative patterns obtained with Gradient × Input. Specifically, while NWP spread decreases moving from early to late layers (i.e., decreasing AUC), the opposite holds for BA attributions. Overall, BA shows substantially higher spread than NWP, indicating that BA depends on more distributed semantic and discourse information integrated across the context, and confirming that this effect is robust across attribution methods.

E.3 FEATURE-BASED ANALYSIS

Figure ~~??~~ 17 presents the percentage of words in each category that are important exclusively for BA, exclusively for NWP, or shared by both, evaluated at three attribution thresholds ($t = 10\%, 60\%, 80\%$). Consistent with prior findings Oh & Schuler (2023) and with ~~GXI~~ Gradient × Input results, NWP shows a clear bias toward syntactic features across all thresholds. In contrast, BA exhibits a more balanced distribution: while syntactic cues remain important Oota et al. (2023b), a substantial proportion of its attribution falls on semantic and discourse-level features. This trend also appears in the intersection set (i.e., words important for both tasks), indicating that NWP partially relies on similar types of words to BA.

E.4 POSITIONAL PATTERNS ANALYSIS

Figures ~~?? and ?? show the distance distribution~~ 18, 19, 20, and 21 compare the distance distributions of top-attributed words ~~under IG for both BA and NWP at two~~ obtained with Integrated Gradients and SmoothGrad at two attribution thresholds (10% and 60%)~~for Llama3.2-1B and Gemma-2B, respectively. IG is computed using $m = 20$ interpolation steps along the path from the baseline to the input, as described in Appendix A.3.~~. Across methods, the core positional trends remain stable: NWP consistently exhibits a pronounced recency peak and, at higher thresholds,

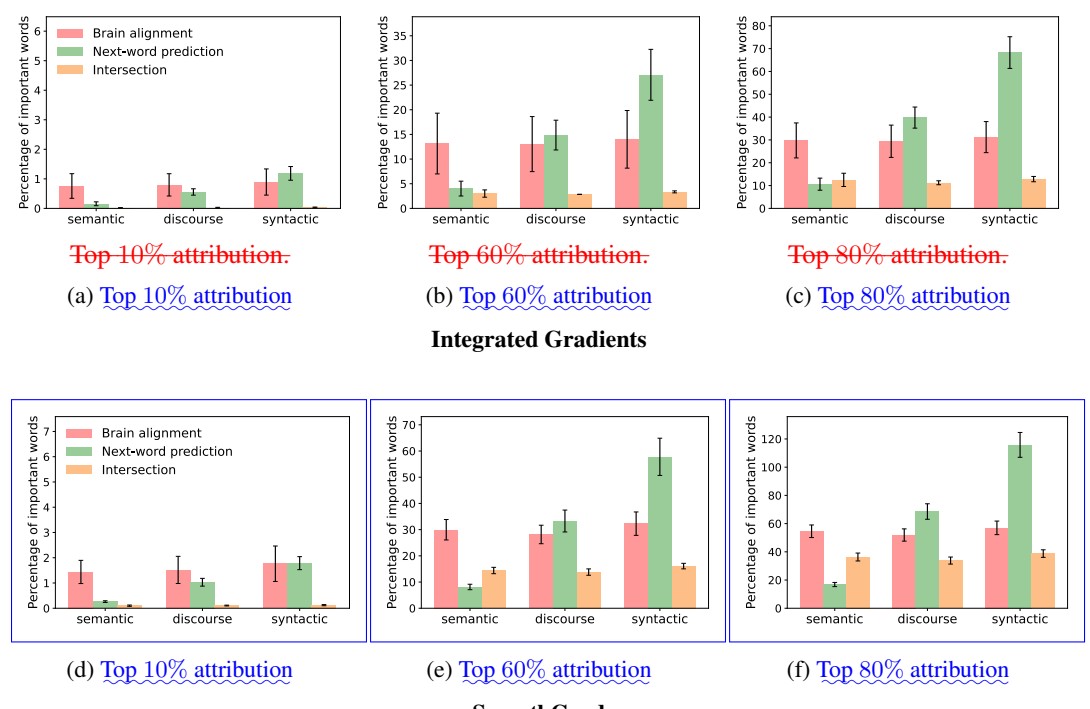

(a) Top 10% attribution  (b) Top 60% attribution  (c) Top 80% attribution

**Integrated Gradients**

(d) Top 10% attribution  (e) Top 60% attribution  (f) Top 80% attribution

**SmoothGrad**

Figure 17: Distribution of top-attributed words across linguistic feature categories (semantic, discourse, syntactic) for brain-alignment-only, next-word-prediction-only, and ~~for~~ words prioritized by both. Results are averaged across layers, subjects, and models, with standard errors across models. We show results for the top 10%, 60%, and 80% ~~of~~ attribution ~~scores~~thresholds. At ~~lower low~~ thresholds, NWP prioritizes syntactic features, while brain alignment emphasizes semantic and discourse-level ~~content~~information. As the threshold increases, ~~the~~ overlap ~~between tasks~~ grows, but brain alignment consistently favors meaning-oriented features.

a secondary primacy peak, whereas BA shows a broader recency focus and a more distributed integration of contextual information.

~~The observed patterns are consistent with those found using GXI. At both thresholds and for both models, NWP shows a bimodal distribution, with strong recency and primacy bias.~~ For Llama3.2-1B, ~~BA displays the same oscillatory and distributed profile we observed with GXI~~Integrated Gradients reproduces the oscillatory attribution pattern observed with Gradient × Input, particularly at ~~the~~ 60% ~~. This supports the conclusion that~~ threshold, indicating that the effect is not specific to a single gradient method. Under SmoothGrad, however, these oscillations largely disappear, yielding a smoother positional profile. This behavior is expected: SmoothGrad averages gradients over multiple noisy perturbations and therefore suppresses fine-grained, high-frequency fluctuations while preserving stable, coarse-scale structure. The disappearance of oscillations under SmoothGrad thus suggests that the effect reflects a stimulus-dependent, high-frequency interaction rather than a global architectural property of the model, confirming the hypothesis we made in the main paper.

Together, these results confirm that the main BA–NWP positional distinctions are robust across attribution methods, while the oscillations specific to Llama3.2-1B ~~integrates context in a structurally distinct manner for brain prediction, and that this behavior is robust to the choice of attribution method. On the contrary, Gemma-2B does not show any oscillatory behavior, and maintains a broader recency peak and less pronounced primacy bias~~are sensitive to smoothing, consistent with their fine-scale and stimulus-dependent nature.

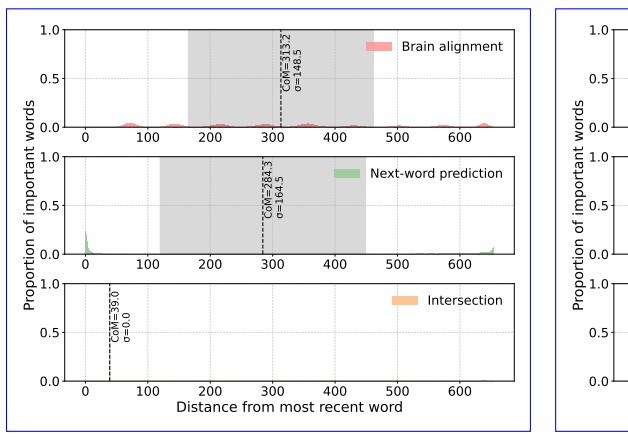 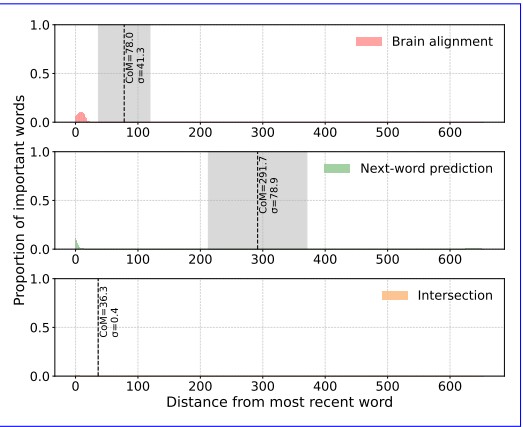

(a) ~~Top 10% attribution~~Llama3.2-1B.          (b) ~~Top 60% attribution~~Gemma-2B.

Figure 18: Distribution of top-attributed words (top 10% attribution) by distance from the most recent word ~~in the context for Llama3.2-1B,~~ using Integrated Gradients. ~~Results are shown for brain alignment~~For each model, ~~next-word prediction~~we plot the proportion of important words located at each distance bin, comparing BA and ~~their intersection~~ NWP. Both BA and NWP show a strong recency bias, with top-attributed words placed at $t = 10\%, 60\%$very low distances from the most recent word.

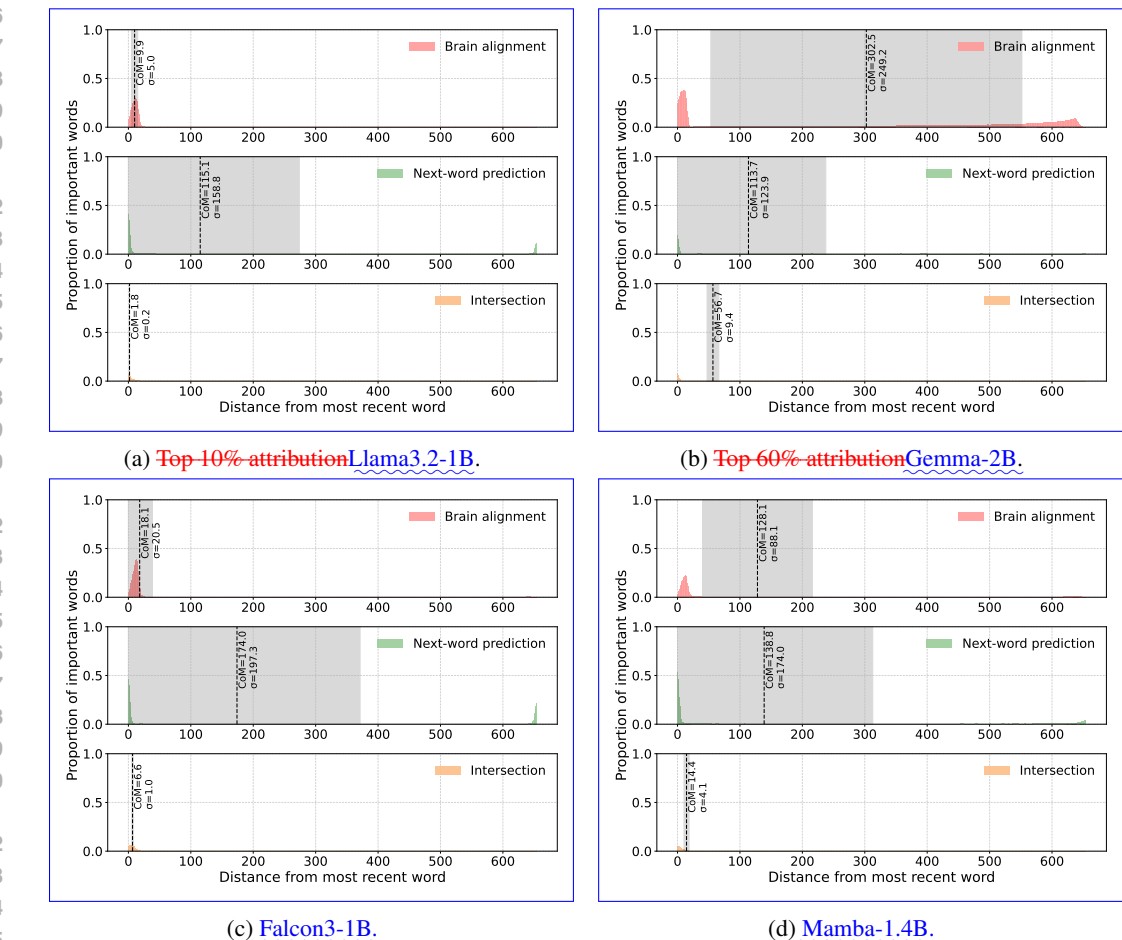

(a) ~~Top 10% attribution~~Llama3.2-1B.

(b) ~~Top 60% attribution~~Gemma-2B.

(c) Falcon3-1B.

(d) Mamba-1.4B.

Figure 19: Distribution of top-attributed words (top 10% attribution) by distance from the most recent word ~~in the context for Gemma-2B,~~ using ~~Integrated Gradients~~SmoothGrad. ~~Results are shown for brain alignment~~For each model, ~~next-word prediction~~we plot the proportion of important words located at each distance bin, comparing BA and ~~their intersection~~NWP. Both BA and NWP show a strong recency bias, with top-attributed words placed at ~~$t = 10\%, 60\%$~~very low distances from the most recent word.

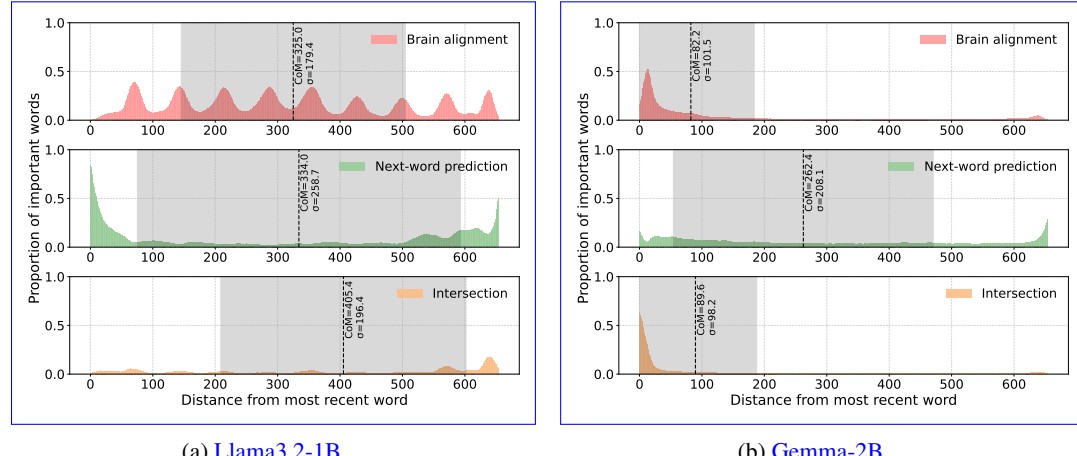

(a) Llama3.2-1B.    (b) Gemma-2B.

Figure 20: Distribution of top-attributed words (top 10% attribution) by distance from the most recent word using Integrated Gradients. For each model, we plot the proportion of important words located at each distance bin, comparing BA and NWP. Both BA and NWP show a strong recency bias, with top-attributed words placed at very low distances from the most recent word.

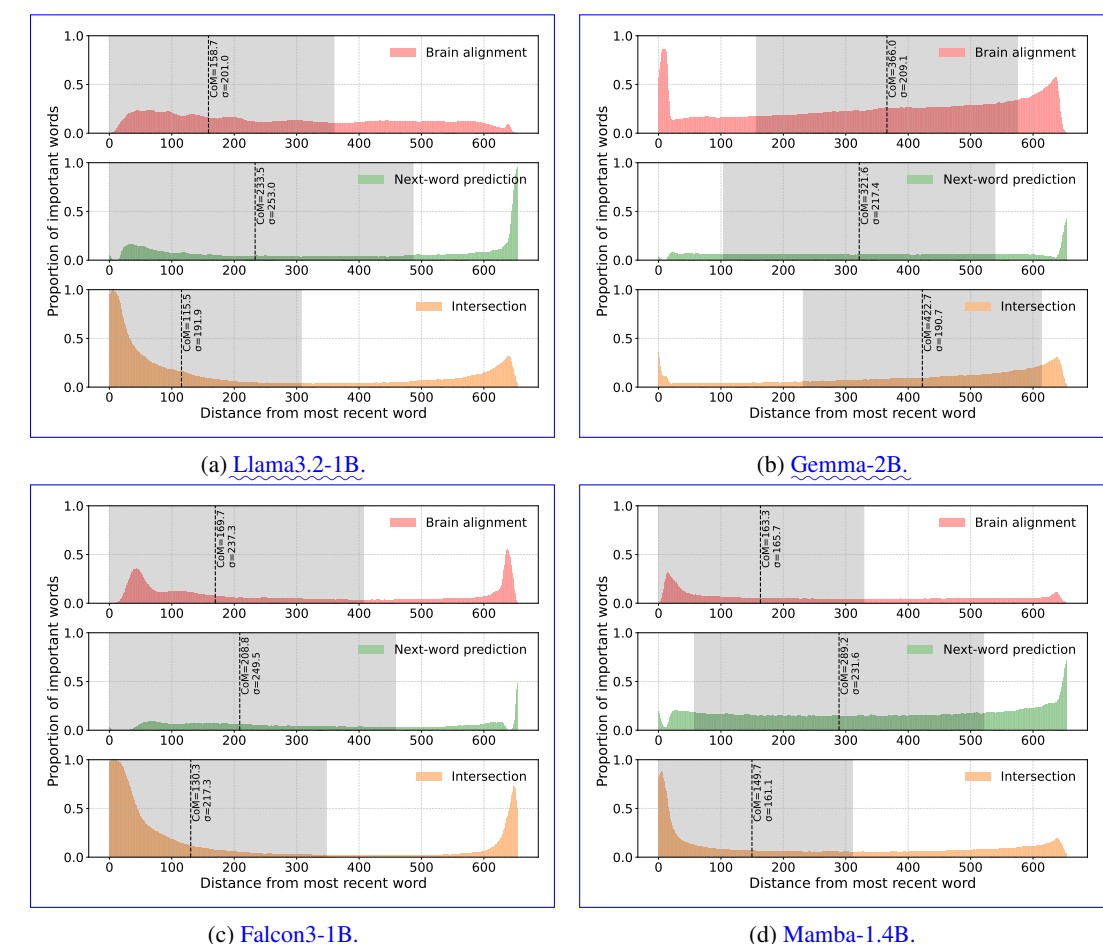

(a) Llama3.2-1B.

(b) Gemma-2B.

(c) Falcon3-1B.

(d) Mamba-1.4B.

Figure 21: Distribution of top-attributed words (top 10% attribution) by distance from the most recent word using SmoothGrad. For each model, we plot the proportion of important words located at each distance bin, comparing BA and NWP. Both BA and NWP show a strong recency bias, with top-attributed words placed at very low distances from the most recent word.

## F  FULL ATTRIBUTION DISTRIBUTION RESULTS

In this section, we provide attribution distribution results for all models, with plots showing the position of top-attributed words relative to the most recent token in the input context. These analyses allow us to assess how BA and NWP differ in terms of the contextual positions they prioritize, and whether these patterns are consistent across architectures and attribution thresholds.

Figure 22 shows the proportion of top-attributed words (top 60% cumulative attribution) located at each distance from the most recent word in the context for Falcon3-1B, Gemma-2B, and Zamba2-1.2B. The dashed lines in each plot represent the center of mass (CoM), computed by considering the top-attributed words. Across models, NWP consistently shows a recency bias, with a sharp peak at low-distance (recent) positions, and an even higher primacy bias, with most attribution mass concentrated at high-distance words. Similarly, BA also shows a strong recency bias, although with a much broader peak (i.e., high-attribution words are present at a higher number of recent positions). In contrast, the primacy bias is much less pronounced. The strong primacy bias for NWP is also the main reason for the higher associated CoM, as high-distance top-attributed words pull it further away from the most recent word.

To evaluate the consistency of our findings across subjects, we plot the attribution distributions for each of the 8 subjects for both Llama3.2-1B and Mamba-1.4B at the 60% attribution threshold. As shown in Figure 23 and Figure 24, the overall trends observed at the group level are highly con-

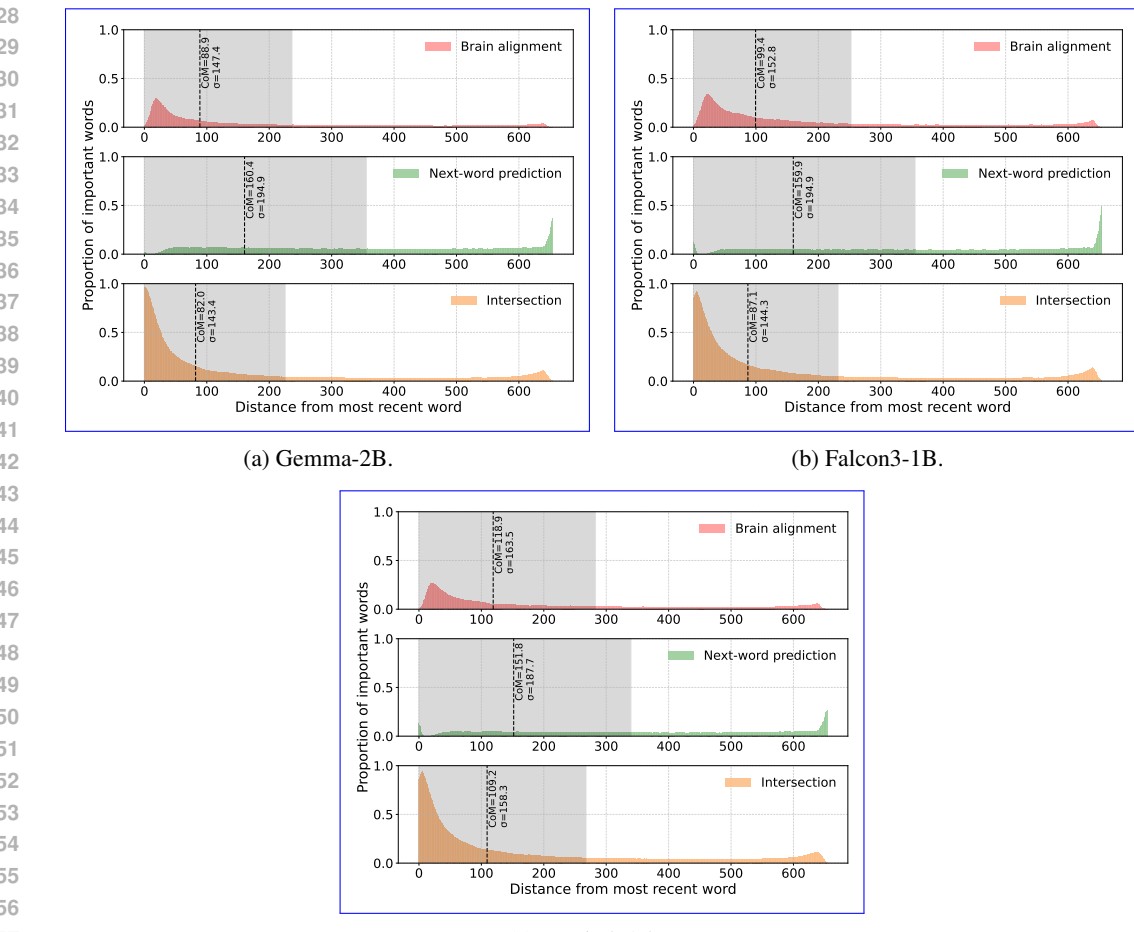

(a) Gemma-2B.

(b) Falcon3-1B.

(c) Zamba2-1.2B.

Figure 22: Distribution of top-attributed words (top 60% attribution) by distance from the most recent word in the context. For each model, we plot the proportion of important words located at each distance bin, comparing brain alignment (BA) and next-word prediction (NWP). NWP shows a strong recency bias, while BA often emphasizes earlier or more distributed words.

sistent across subjects. In particular, NWP consistently shows a strong recency and primacy bias. BA, instead, emphasizes a broader distribution over more recent tokens with minimal primacy bias for Mamba-1.4 and the peculiar oscillatory pattern for Llama3.2-1B. The subject-specific curves exhibit similar shapes and centers of mass, indicating low inter-subject variability and supporting the robustness of the main effects. These results validate our choice to report mean attribution distributions across subjects in the main text and reinforce the interpretability of the observed differences between BA and NWP.

### F.1 ATTRIBUTION DISTRIBUTION FOR $t = 10\%$

Figure 25 shows the same analysis at a lower attribution threshold of $10\%$. These plots focus on the most influential words per context, offering a finer-grained view of attribution prioritization. While NWP continues to show a sharp peak near the end of the context (strong recency), BA exhibits a less sharply peaked but still recency-focused distribution in most models.

The CoM for both BA and NWP is generally closer to the recent words than in the $60\%$ case, indicating that when only the most highly attributed words are considered, BA and NWP become more similar in their positional focus. Still, notable differences remain between Llama3.2-1B and all other models, as the oscillatory distribution pattern for BA already emerges at $t = 10\%$.

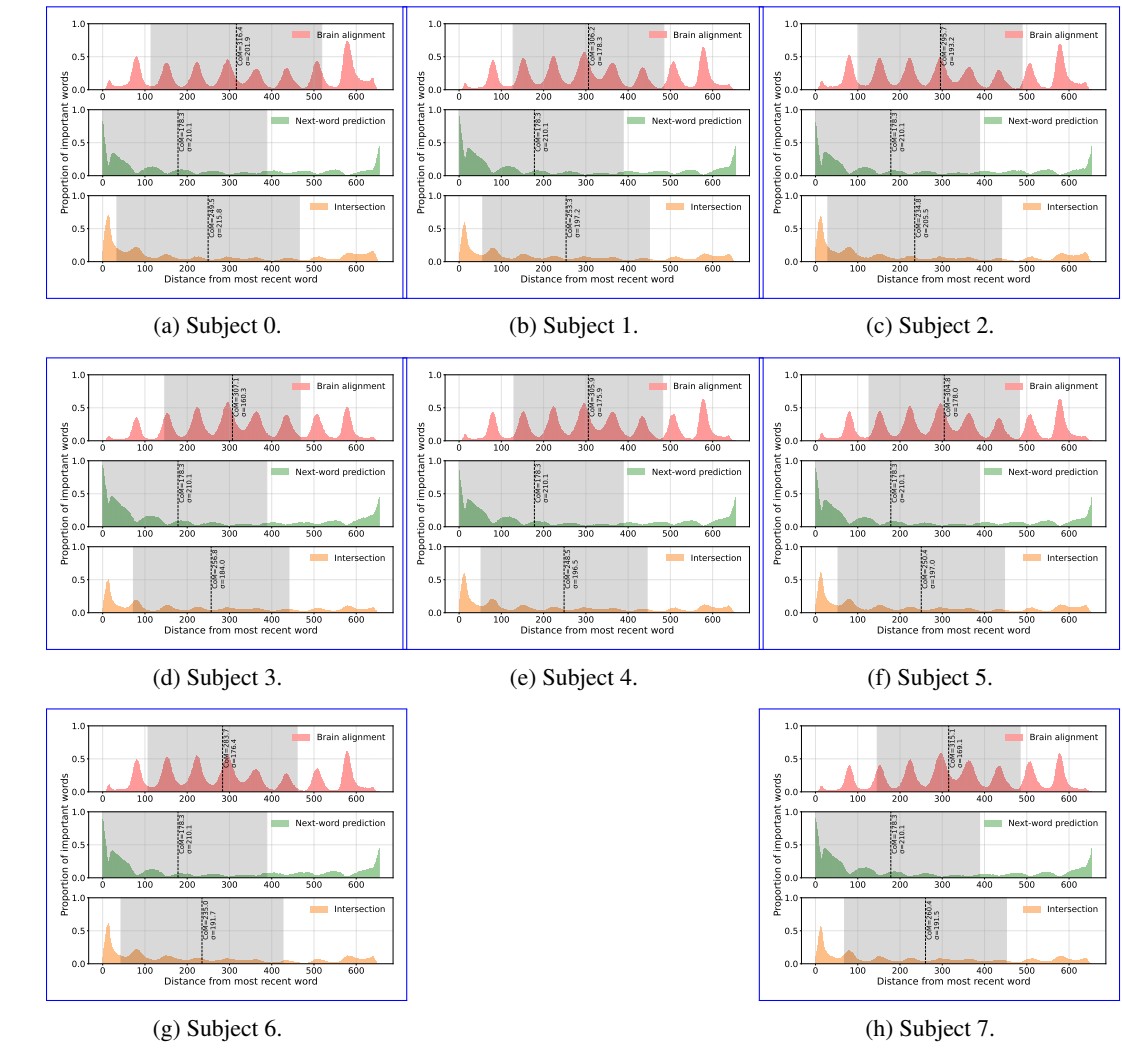

(a) Subject 0.  (b) Subject 1.  (c) Subject 2.

(d) Subject 3.  (e) Subject 4.  (f) Subject 5.

(g) Subject 6.  (h) Subject 7.

Figure 23: Distribution of top-attributed words (top 60% cumulative attribution) by distance from the most recent word in the context, shown separately for each of the 8 subjects for Llama3.2-1B. Each plot compares brain alignment and next-word prediction. The overall shape and center of mass of the attribution distributions are consistent across subjects, indicating low inter-subject variability. This supports the robustness of the observed task-specific attribution differences.

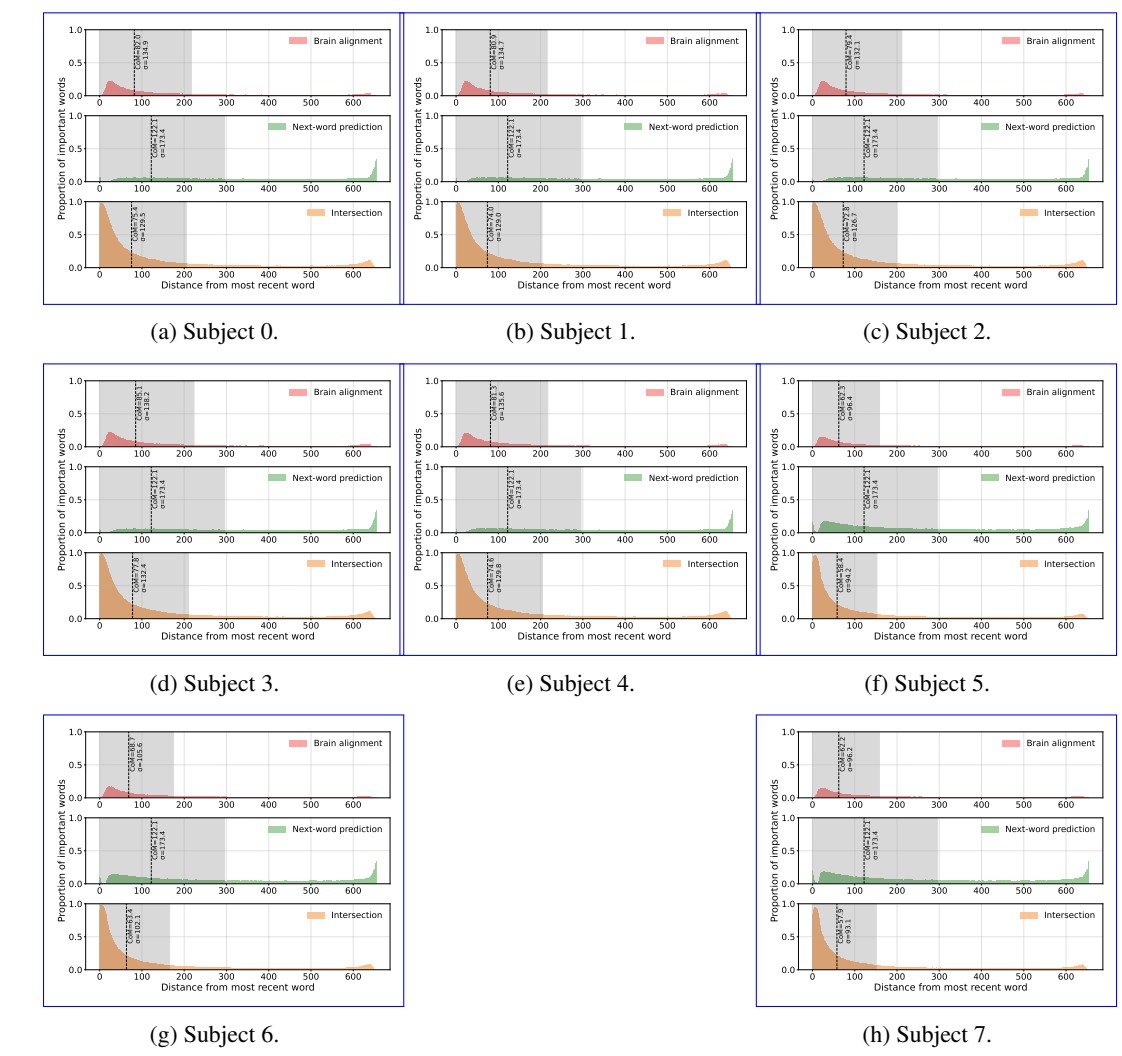

(a) Subject 0.     (b) Subject 1.     (c) Subject 2.

(d) Subject 3.     (e) Subject 4.     (f) Subject 5.

(g) Subject 6.                (h) Subject 7.

Figure 24: Distribution of top-attributed words (top 60% cumulative attribution) by distance from the most recent word in the context, shown for each of the 8 subjects for Mamba-1.4B. Across all subjects, next-word prediction shows a strong recency and primacy bias, while brain alignment attribution is more concentrated around the recent context and more broadly distributed over recent words. The similarity in curves across subjects confirms that these effects are consistent at the individual level.

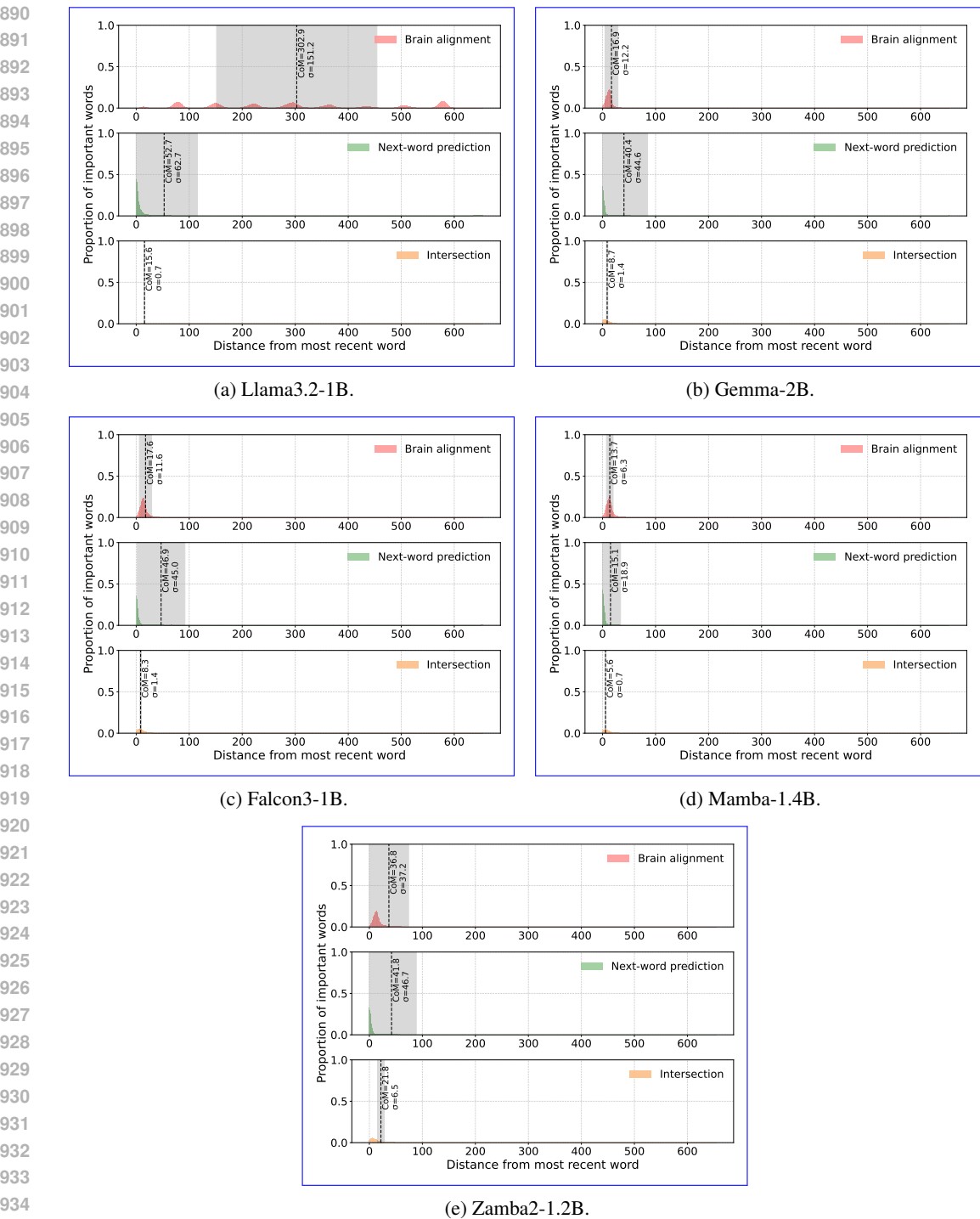

(a) Llama3.2-1B.

(b) Gemma-2B.

(c) Falcon3-1B.

(d) Mamba-1.4B.

(e) Zamba2-1.2B.

Figure 25: Distribution of top-attributed words (top 10% attribution) by distance from the most recent word in the context. For each model, we plot the proportion of important words located at each distance bin, comparing brain alignment (BA) and next-word prediction (NWP). NWP shows a strong recency bias, while BA often emphasizes earlier or more distributed words.

## G QUANTITATIVE RESULTS FOR BRAIN ALIGNMENT

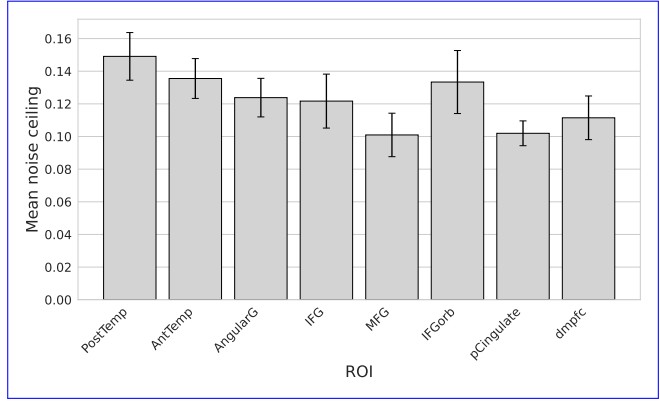

(a) Noise ceiling.

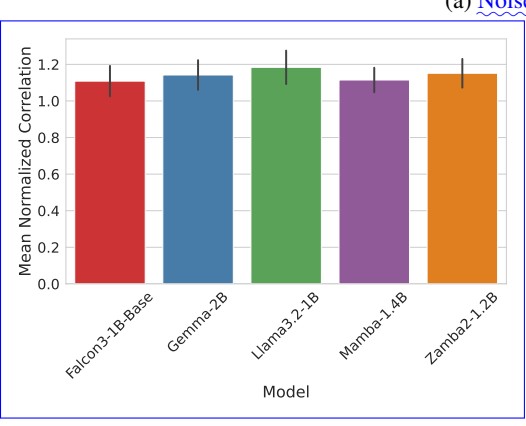

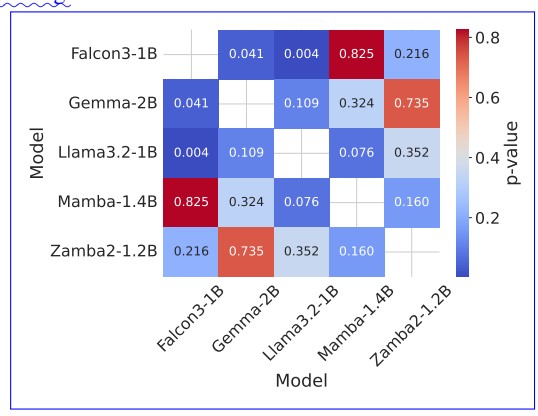

(b) Model-wise mean correlation.

(c) Pair-wise significance heatmap.

Figure 26: Model-wise brain alignment performance. (a) Noise ceiling computed using cross-subject prediction accuracy (Schrimpf et al., 2021). (b) Mean normalized correlation between true and predicted brain activity across models. We report the mean across voxels, layers, and subjects, with standard error across subjects. (b) Pairwise significance heatmap showing $p$-values from paired t-tests comparing mean correlation scores between models.

Figure 26 presents quantitative BA scores for the five LLMs evaluated in this study: Falcon3-1B, Gemma-2B, Llama3.2-1B, Mamba-1.4B, and Zamba2-1.2B. BA is measured as the mean Pearson correlation between predicted and observed fMRI activity, averaged across all voxels, layers, and subjects.

To account for measurement noise inherent in biological data and to obtain a more reliable estimate of model performance, we compute the noise ceiling following Schrimpf et al. (2021). Specifically, we compute cross-subject prediction performance, i.e. how much of a subject's brain response can be predicted using data from other subjects alone, and report results in Figure 26a.

Figure 26b shows a bar plot of the model-wise mean normalized correlation scores, with standard error bars reflecting variability across subjects. Among the evaluated models, Llama3.2-1B achieves the highest average BA, followed by Zamba2-1.2B and Gemma-2B. Falcon3-1B and Mamba-1.4B exhibit lower alignment scores, suggesting reduced ability to capture brain-relevant representations.

Figure 26c, instead, shows a pairwise significance heatmap, where each cell reports the $p$-value of a two-sided paired t-test comparing voxel-wise normalized correlations between two models across subjects. Both Llama3.2-1B and Gemma-2B show significantly higher alignment than Falcon3-1B ($p < 0.05$), with Llama3.2-1B also showing low $p$-values in comparisons with Mamba-1.4B, suggesting consistently stronger alignment overall. No significant differences are observed be-

tween Falcon3-1B, Mamba-1.4B, and Zamba2-1.2B, indicating comparable performance among these models within the margin of statistical uncertainty.

## H  POSITIONAL PATTERNS ANALYSIS ON QWEN2-1.5B

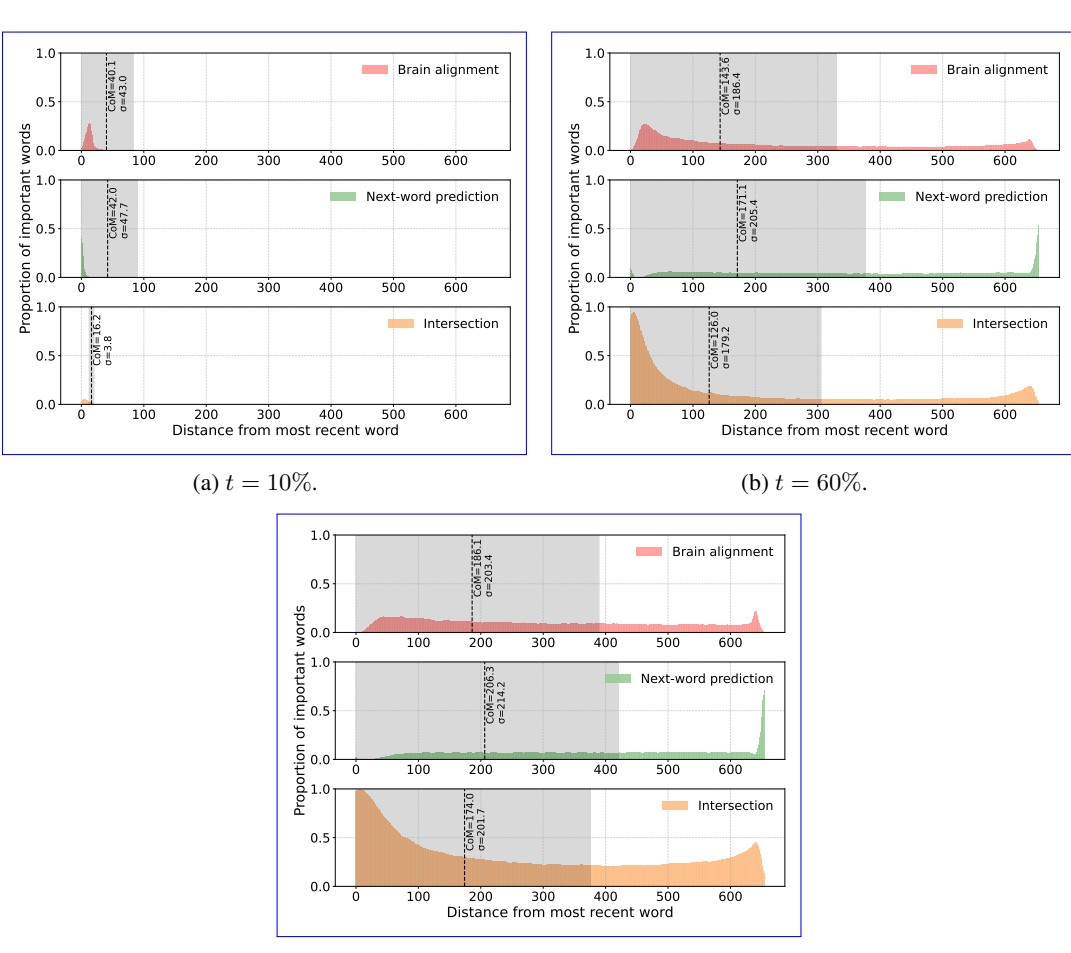

(a) $t = 10\%$.            (b) $t = 60\%$.

(c) $t = 80\%$.

Figure 27: Distribution of top-attributed words by distance from the most recent word in the context for Qwen2-1.5B, shown at thresholds $t = 10, 60, 80\%$. Each plot compares brain alignment (BA) and NWP. Across thresholds, NWP displays a sharp recency bias and a secondary primacy peak, while BA shows a broader recency distribution. Unlike Llama3.2-1B on the Harry Potter dataset, Qwen2-1.5B does not exhibit oscillatory attribution patterns, instead maintaining smooth and monotonic profiles.

We analyze positional attribution patterns for Qwen2-1.5B, a transformer architecture that shares several design features with Llama3.2-1B, including rotary position embeddings (RoPE), grouped-query attention (GQA), FlashAttention2, and similar quantization strategies. Figures 27 shows the distribution of top-attributed words at thresholds $t = 10, 60, 80\%$, plotted as a function of distance from the most recent word in the input context.

Across all thresholds, Qwen2-1.5B exhibits the canonical task-dependent positional profiles observed in other models. NWP consistently displays a sharp recency bias, with most attribution mass concentrated on the last few tokens, and a secondary primacy peak becoming more apparent at higher thresholds. BA, by contrast, shows a broader recency distribution, with important words spread more evenly across the recent context window. Importantly, unlike Llama3.2-1B, Qwen2-1.5B does not display oscillatory attribution patterns for BA. Instead, its attribution profile remains smooth across thresholds. This suggests that oscillatory behavior is not a necessary consequence

of shared architectural components (RoPE, GQA, or FlashAttention2), but rather may emerge from interactions between architecture and specific input statistics, as confirmed by the absence of oscillations when considering shorter contexts or on the ~~MRH~~ Moth Radio Hour dataset. Together, these results reinforce the robustness of the general trends: NWP dominated by sharp recency and primacy, and BA characterized by smoother and broader recency.

# I ATTRIBUTION ANALYSIS WITH SHORTER CONTEXTS

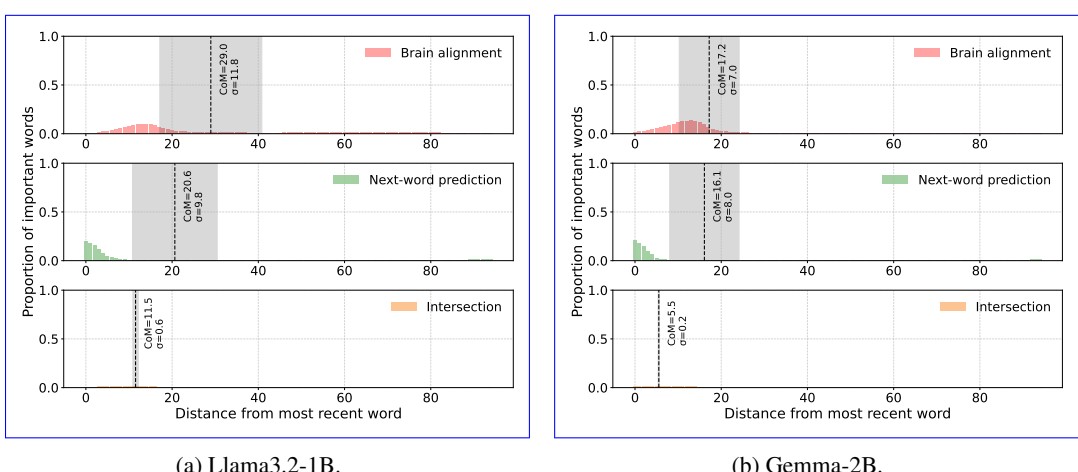

(a) Llama3.2-1B.    (b) Gemma-2B.

Figure 28: Distribution of top-attributed words (top 10% attribution) by distance from the most recent word in the context, using 80-word contexts. For each model, we plot the proportion of important words located at each distance bin, comparing brain alignment (BA) and next-word prediction (NWP). NWP shows a strong recency bias, while BA often emphasizes earlier or more distributed words.

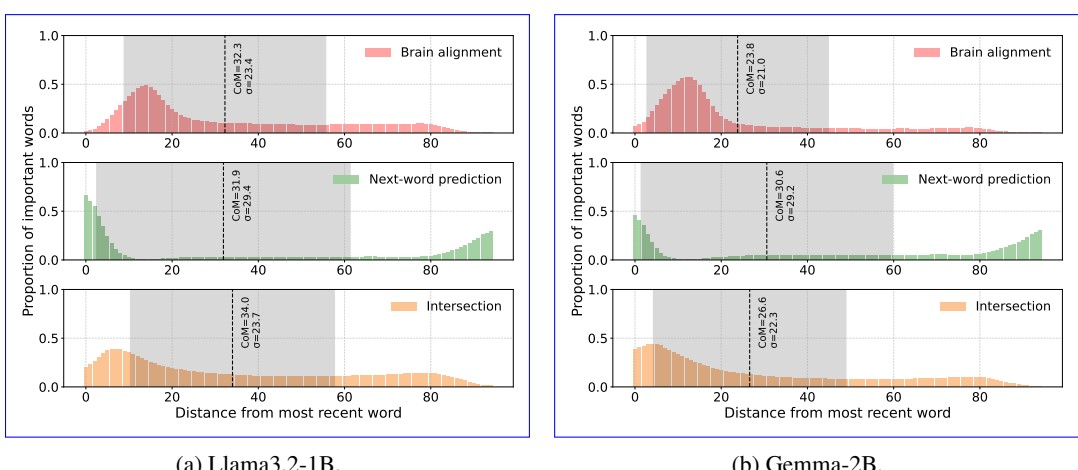

(a) Llama3.2-1B.    (b) Gemma-2B.

Figure 29: Distribution of top-attributed words (top 60% attribution) by distance from the most recent word in the context, using 80-word contexts. For each model, we plot the proportion of important words located at each distance bin, comparing brain alignment (BA) and next-word prediction (NWP). NWP shows a strong recency bias, while BA often emphasizes earlier or more distributed words.

~~Llama3.2-1B. Gemma-2B.~~

~~Distribution of top-attributed words (top 80% attribution) by distance from the most recent word in the context, using 80-word contexts. For each model, we plot the proportion of important words~~

~~located at each distance bin, comparing brain alignment (BA) and next-word prediction (NWP). NWP shows a strong recency bias, while BA often emphasizes earlier or more distributed words.~~

To assess robustness to context length, we repeated our full pipeline using 80-word contexts (an 8× reduction from the 640-word setting used in all other experiments).

**Attribution profiles are maintained.** Figures 28–~~??~~ 29 report attribution distributions for Llama3.2-1B and Gemma-2B obtained using the shorter contexts. The overall shape of the attribution distributions remains consistent with the longer-context results: BA maintains a smoother profile with a broad recency peak at ~12 words, while NWP preserves its sharp bimodal structure: a strong spike on the most recent 5 words and a second, smaller primacy bump at the farthest positions (75–80 words), underscoring its heavy reliance on sentence-edges (Liu et al., 2024). This confirms these positional biases are not an artifact of long contexts.

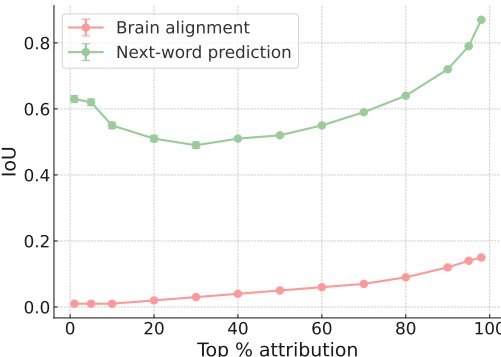

Figure 30: IoU of top-attributed words between long (640-word) and short (80-word) contexts as a function of attribution mass threshold $t$. NWP shows high robustness across window sizes, while BA exhibits low overlap, reflecting its more distributed and flexible use of context.

**Important words overlap between short and long contexts.** We compared attributions across long and short contexts to understand whether the same words are identified as important in both settings, by computing Spearman's $\rho$ for rank agreement (restricted to the last 80 words of the 640-word contexts) and IoU of the top-mass words (shown in Figure 30. NWP proves highly robust, with strong rank correlation ($\rho \approx 0.8$) and substantial overlap of important words ($IoU > 0.5$ across all thresholds). In contrast, BA shows only moderate rank agreement ($\rho \approx 0.44$) and very limited overlap ($IoU < 0.1$ for $t < 80\%$), except at very large thresholds.

These findings highlight that NWP relies on a narrow, stable set of recent words, while BA distributes attribution more flexibly across context, leading to lower robustness in top-word overlap across window sizes.

## J  EXPERIMENTS COMPUTE RESOURCES

All experiments were conducted on a single NVIDIA H100 Tensor Core GPU (80GB, Hopper architecture). Tables 3–5 summarize the time and memory requirements for each task and model. To estimate total compute usage, we sum the runtimes of all major experimental stages across the five evaluated models:

- **LLM representation extraction**: ≈10 minutes × 5 models × 2 datasets = ≈2 hours total
- **Brain alignment training**: ≈219 hours total
- **~~GXI~~ Gradient × Input attributions**: ≈1501 hours total
- **~~IG~~ Integrated Gradients attribution**: ≈329 hours total
- **NWP attribution**: ≈3.6 hours total

While attribution is computationally intensive, we mitigate resource demands by limiting our study to 1–2B parameter models and using a single H100 GPU per task. These design choices balance tractability with representational fidelity and enable detailed interpretability analyses within practical runtime constraints.

Table 3: Per-task compute time and peak memory usage across models on the Harry Potter dataset.

| Task | Model | Time (dd:hh:mm:ss) | Peak Memory (GB) |
|---|---|---|---|
| Brain alignment training | Llama3.2-1B | 00:04:08:16 | 3.16 |
| | Gemma-2B | 00:04:47:19 | 3.08 |
| | Falcon3-1B | 00:05:42:46 | 3.39 |
| | Zamba2-1.2B | 00:10:24:00 | 3.66 |
| | Mamba-1.4B | 00:12:50:23 | 3.86 |
| | Qwen2-1.5B | 00:05:46:31 | 3.46 |
| Gradient × Input attribution | Llama3.2-1B | 01:14:00:00 | 2.53 |
| | Gemma-2B | 08:16:03:12 | 2.45 |
| | Falcon3-1B | 07:10:08:01 | 2.53 |
| | Zamba2-1.2B | 13:03:12:00 | 2.63 |
| | Mamba-1.4B | 11:12:00:00 | 2.53 |
| | Qwen2-1.5B | 00:15:36:34 | 2.47 |
| Integrated Gradients attribution | Llama3.2-1B | 13:13:43:01 | 2.53 |
| | Gemma-2B | 00:03:34:39 | 2.51 |
| NWP attribution | Llama3.2-1B | 00:00:22:58 | 2.18 |
| | Gemma-2B | 00:00:08:26 | 2.16 |
| | Falcon3-1B | 00:00:06:07 | 2.17 |
| | Zamba2-1.2B | 00:00:14:20 | 2.26 |
| | Mamba-1.4B | 00:00:07:38 | 2.18 |
| | Qwen2-1.5B | 00:00:05:47 | 2.22 |

Table 4: Per-task compute time and peak memory usage across models on the Moth Radio Hour dataset.

| Task | Model | Time (dd:hh:mm:ss) | Peak Memory (GB) |
|---|---|---|---|
| Brain alignment training | Llama3.2-1B | 00:08:44:08 | 34.49 |
| | Gemma-2B | 00:10:19:46 | 36.05 |
| | Falcon3-1B | 01:11:56:51 | 40.98 |
| | Zamba2-1.2B | 02:00:45:34 | 36.24 |
| | Mamba-1.4B | 02:11:36:54 | 34.78 |
| Gradient × Input attribution | Llama3.2-1B | 00:12:58:21 | 28.22 |
| | Gemma-2B | 00:14:52:05 | 28.07 |
| | Falcon3-1B | 03:28:07:06 | 28.15 |
| | Zamba2-1.2B | 08:15:36:44 | 27.85 |
| | Mamba-1.4B | 04:19:02:20 | 27.63 |
| NWP attribution | Llama3.2-1B | 00:00:08:00 | 26.87 |
| | Gemma-2B | 00:00:10:55 | 26.86 |
| | Falcon3-1B | 00:00:18:50 | 26.36 |
| | Zamba2-1.2B | 00:00:59:26 | 26.46 |
| | Mamba-1.4B | 00:00:21:07 | 26.38 |

## K LLM USAGE STATEMENT

We used LLMs solely for grammar checking and minor wording improvements. All research ideas, analyses, experiments, and results are entirely our own.

Table 5: Per-task compute time and peak memory usage across models for experiments with reduced context length.

| Task | Model | Time (dd:hh:mm:ss) | Peak Memory (GB) |
|---|---|---|---|
| Brain alignment training | Llama3.2-1B | 00:05:40:33 | 3.01 |
| | Gemma-2B | 00:06:13:33 | 2.99 |
| Gradient × Input attribution | Llama3.2-1B | 00:09:57:10 | 2.59 |
| | Gemma-2B | 00:09:52:24 | 2.57 |
| NWP attribution | Llama3.2-1B | 00:00:22:58 | 2.18 |
| | Gemma-2B | 00:00:08:26 | 2.16 |

