# OpenReview forum: "Fine-grained Analysis of Brain-LLM Alignment through Input Attribution"
_ICLR.cc/2026/Conference — Submitted to ICLR 2026_

### Official Review · Reviewer_4jvo · 2025-10-29

**Soundness:** 2
**Presentation:** 2
**Contribution:** 2
**Rating:** 2
**Confidence:** 4

**Summary:**

The paper proposes a unified input attribution framework to compare how large language models (LLMs) align with human brain activity (Brain Alignment, BA) versus next-word prediction (NWP). Using gradient-based word-level attributions, the authors show that BA and NWP rely on distinct linguistic features: NWP emphasizes syntactic and short-range cues, while BA depends more on semantic and discourse information. The study provides a fine-grained, interpretable view of LLM-brain alignment and highlights differences between predictive and comprehension-related representations.

**Strengths:**

1. Novel use of quantitative attribution metrics.

The paper introduces and adapts two interpretable metrics, Intersection over Union (IoU) and Center of Mass (CoM), to quantify the overlap and positional tendencies of important words in Brain Alignment (BA) and Next-Word Prediction (NWP) tasks. This design enables a clear, measurable comparison of attribution patterns across tasks and represents a well-motivated methodological contribution.

2. Inclusion of diverse model architectures.

The study incorporates state-space models (SSMs) alongside traditional Transformer architectures in its analysis. This cross-architecture comparison is relatively new in the brain–language modeling literature and broadens the generality of the findings, demonstrating that the observed attribution trends are consistent across distinct model families.

**Weaknesses:**

1. **Limited conceptual novelty.**

The main weakness of the paper lies in the limited originality of its findings. The preference of brain alignment (BA) tasks for semantic information has been recognized in neuroscience, and the syntactic bias of next-word prediction (NWP) has also been observed in NLP studies. While the attribution-based comparison provides finer detail, it largely confirms existing conclusions rather than extending them conceptually.

2. **Incompleteness of the CoM metric.**

The Center-of-Mass (CoM) index captures only the mean positional tendency of attributions but ignores their degree of dispersion. Visual inspection suggests that in some results, word attributions are highly scattered or concentrated. Adding a complementary measure (e.g., variance or entropy of attribution positions) would provide a more comprehensive characterization of positional patterns.

3. **Gradient-based attribution limitations.**

The study relies solely on gradient-based attribution (Gradient $\times$ Input and Integrated Gradients), which are inherently sensitive to local nonlinearities and model saturation. Although Integrated Gradients mitigates these issues, it does not fully resolve them. Incorporating additional attribution approaches, such as perturbation-based or ensemble methods (e.g., SmoothGrad), would improve robustness and better justify the use of the term "attribution".

**Questions:**

1. In several figures (e.g., Figure 5, Mamba), the NWP task shows a clear rise in attribution for tokens beyond position 600, yet the reported CoM value remains below 200. Does this indicate that the CoM metric is insensitive to high but localized attribution peaks in distant contexts?

2. The current analysis focuses on comparing attribution distributions but does not address the deeper question that the field increasingly finds important—how strongly BA and NWP are related, and what mechanisms underlie their correlation. Since the authors already measure both tasks, explicitly quantifying and explaining this relationship would make the contribution more meaningful.

---

> ### Author Response · Authors · 2025-11-24
>
> # Limited conceptual novelty
> We thank the reviewer for raising this important concern. While prior work has indeed suggested that BA is more sensitive to semantic information and that NWP often relies on syntactic cues, our goal is not simply to restate these broad tendencies. Rather, we aim to provide a fine-grained characterization of how each task draws on information in the input. Existing evidence comes primarily from coarse perturbations of inputs or model representations, which make it difficult to determine which specific words drive alignment or how these words are distributed across the context.
>
> Our attribution framework addresses this gap by offering word-level, end-to-end explanations, allowing us to measure (i) where relevant information is located in the sequence, (ii) how broadly each task uses context, and (iii) how linguistic properties contribute in a task-dependent manner. This reveals new functional distinctions, including differences in recency structure and attribution spread, that were not accessible with previous methods. In this sense, our work provides a higher-resolution view of how BA and NWP extract information from the same underlying representations.
>
> Overall, our main contributions are methodological: we use input attribution to explain brain-LLM alignment and adapt quantitative metrics to analyze attribution patterns. The BA-NWP comparison serves as a case study illustrating the types of insights enabled by this framework. However, the results both replicate known patterns, supporting the validity of our approach, and reveal new differences in how BA and NWP functionally use different parts of the input context, providing novel conceptual insights into the relationship between the two tasks.
>
> # Deeper question of how strongly NWP and BA are related remains unanswered
> We thank the reviewer for raising this important point. Our goal is indeed to characterize how BA and NWP relate, and our attribution-based comparison directly targets this relationship. By identifying which input words are functionally used by each task, and quantifying their overlap, positional biases, linguistic properties, and spread, we measure the degree and structure of alignment between BA and NWP at a much finer resolution than prior work. The consistent findings across models and subjects reveal both where the two tasks rely on shared representational content and where they diverge in functional usage.
>
> Although we do not aim to measure “overall task correlation’’ (e.g., correlation between BA $r^2$ and NWP accuracy), our analyses already address the mechanism underlying the partial relationship: both tasks draw on the same underlying LLM representations, but they extract different types of information from different parts of the context. In the revision, we clarify this framing in the introduction and explicitly state that the attribution comparison is our measure of the BA–NWP relationship: it reveals how the two tasks differentially exploit the same representational substrate.
>
> # Incompleteness of the CoM metric
> We thank the reviewer for this constructive suggestion, which has strengthened our analysis and clarified important differences between BA and NWP attribution profiles. We fully agree that the CoM captures only the average positional tendency of attributions and does not reflect how narrowly or broadly they are distributed across the context.
> In the revised manuscript, we have therefore added a complementary attribution-spread metric (Section 3.4), which computes a weighted positional standard deviation of attribution scores, directly quantifying dispersion: low spread indicates tightly clustered recency or primacy peaks, while high spread reveals broad contextual integration or multi-modal attribution patterns.
> We have already updated most of Figures 5, 12, 13, 14, 17–21, and 23–26 to include this metric as a shaded region around the CoM line.
>
> # Limitations of gradient-based attributions
> We appreciate the reviewer’s point. We agree that complementary attribution techniques help increase robustness. We are currently running the full attribution pipeline using SmoothGrad. These results will be incorporated into the revised manuscript within the next few days.
>
> # Question on low CoM, compared to primacy peaks
> We thank the reviewer for this careful observation. The apparent discrepancy arises from the fact that the y-axis in Figure 5 reflects the proportion of important words at each relative position, not the absolute attribution mass. Early in each fold, the extended contexts are shorter (because the model has seen fewer preceding TRs), so positions far from the last word contain fewer total words across examples. Thus, the CoM remains low not because it is insensitive to distant peaks, but because those peaks correspond to fewer actual words relative to the dense cluster of important words near the most recent positions. We will clarify this aspect in the revised manuscript.

---

> > ### Author Response · Authors · 2025-11-28
> >
> > Dear Reviewer 4jvo,
> >
> > We truly appreciate your effort in evaluating our work and your thoughtful suggestions.
> >
> > In response to your insightful comments, we have addressed the issues you highlighted and added new experiments. We uploaded a new revision of the manuscript, with all modifications clearly marked in color to help identify the updates. We believe these revisions significantly contribute to the clarity and completeness of the paper. In the following, we highlight some updates with respect to our previous comment.
> >
> > ## Update: Limitations of gradient-based attributions
> >
> > Thank you again for the helpful suggestion to broaden the attribution analyses. As an update, we have now completed SmoothGrad attribution runs for all models except Zamba2-1.2B, and the results fully support the patterns previously observed with Gradient × Input and Integrated Gradients: BA–NWP overlap remains low across thresholds, layer-wise spread differences replicate robustly, and BA continues to exhibit broader, semantically driven contextual integration.
> >
> > To reflect this expanded analysis, we have updated the manuscript as follows:
> > - Appendix A.3 now includes a description of SmoothGrad and how it is applied in our pipeline.
> > - Appendix E now includes SmoothGrad IoU, positional, and attribution-spread results for Llama3.2-1B, Gemma-2B, Falcon3-1B, and Mamba-1.4B; we will update these figures immediately once the Zamba2-1.2B run completes.
> > - We also revised the main text (in the introduction to §3.3 and in the Results) to explicitly reference SmoothGrad as additional confirmation that our findings are stable across multiple attribution methods.
> >
> > These additions strengthen the methodological robustness of the attribution analyses and further support the reliability of the conclusions.

---

### Official Review · Reviewer_P1Ga · 2025-10-30

**Soundness:** 3
**Presentation:** 3
**Contribution:** 2
**Rating:** 4
**Confidence:** 2

**Summary:**

This paper proposes a new end-to-end attribution framework for studying how large language models align with human brain activity. The authors compute word-level attributions for both brain alignment (BA) (predicting fMRI signals from model embeddings) and next word prediction (NWP), then compare which parts of the input each task depends on.

They introduce 2 complementary measures: intersection-over-union (IoU) and Center-of-Mass (CoM) to quantify overlap and positional bias of attributions. They use the Harry Potter and Moth Radio Hour fMRI datasets and 5 pretrained LLMs between 1B and 2B (Transformers, State-Space Model and Hybrid). They find that:

- BA and NWP rely on largely distinct sets of words, with little overlap for the top-attributed inputs.
- NWP is dominated by syntactic and edge-biased cues (recency and primacy effects).
- BA relies more on semantic and discourse-level information with broader and more distributed recency effects.
- These trends generalize across the 2 datasets and architectures.

The framework provides a quantitative bridge between representational neuroscience and interpretability in LLMs.

**Strengths:**

- Methodological rigor: The mathematical grounding of the framework is solid. The use of IoU and CoM as geometric measures of overlap and temporal focus is elegant and interpretable and the end-to-end gradient-based formulation is technically sound.

- Semantic intepretation of BA: The result that BA emphasizes semantic and discourse cues more than NWP is both intuitive and empirically grounded. It strengthens the argument that the brain's predictive machinery operates at a higher representational level than next-token statistics.

- Potential as a reusable framework: the attribution pipeline could become a standard tool for probing cognitive relevance in multimodal brain-AI comparisons.

The work is carefully executed, but the framework is so promising that it leaves the reader wanting more. Particularly a deeper analysis of the architectural differences and a more thorough exploration of the attribution results.

**Weaknesses:**

- Underdeveloped discussion of model classes: the transformer vs. SSM comparison is one of the most exciting parts, yet the discussion (a few lines around Fig. 5) is minimal. Why does Mamba behave differently? What architectural biases could explain the observed recency differences?

- Attribution: the paper claims to present an end-to-end framework for attribution, but the application is restricted to three categories: semantic, syntactic, and discourse (the latter is not clearly defined in the paper). Given the setup, one would expect masking or perturbation ablations to confirm causal influence. These are absent, except for a brief mention in the appendix.

- Dataset dependence: a major limitation is that the findings are largely based on the Harry Potter reading fMRI dataset, which is extremely narrow in style. The brief extension to The Moth Radio Hour is not enough to claim generality. It would have been particularly interesting to see whether the same patterns hold for non-narrative or multilingual data.

- Clarity and presentation: using the proportion of important words by distance as the core positional plot (in several versions, especially in the appendix) is not necessarily the most effective way to convey the results. Most figures (except Figure 2) are difficult to interpret, and simpler visualizations could make the findings clearer. The paper could also be shortened substantially (currently 34 pages) without losing substance.

I would be happy to revisit my rating should the authors address the points raised in this review.

**Questions:**

1. How exactly do transformers and SSMs differ in their attribution profiles?

2. Which model achieved the highest brain alignment overall and which the lowest? What architectural or representational properties might explain this ranking ?

2.bis. What happened to Llama3.2-1B in Figure 2? Is there an explanation vs. the other models?

3. If BA is more semantic, can this be causally demonstrated by masking high semantic weight words?

4. Why were only 3 categories for attribution grouping? Harry Potter annotations allow richer semantic subtypes. Could the analysis be more granular ?

5. How stable are the attribution patterns across subjects and runs?

6. Can you think of a way this framework handle multimodal (audio, vision) stimuli where tokenization is less discrete ?

---

> ### Author Response · Authors · 2025-11-24
> **Rebuttal to weaknesses**
>
> # Underdeveloped discussion on model classes
> We thank the reviewer for raising this point. We agree that architectural comparisons are important, and we added a dedicated paragraph in the Discussion section. Across all models (Transformers, SSMs, and hybrids) we observe the the same core attribution patterns: low BA-NWP attributions overlap; greater NWP spread in early layers and greater BA spread in middle/late layers; a bimodal NWP distance distribution (recency + primacy) versus a broader BA recency peak. These robust cross-architecture effects suggest that the differences between BA and NWP reflect task-driven functional usage of shared representations rather than architectural idiosyncrasies.
>
> # Llama’s outlier behaviour
> Llama3.2-1B shows lower IoU in Fig. 2, but this does not reflect weaker brain alignment (Llama3.2-1B is the most brain-aligned model on the HP dataset). Instead, its IoU is reduced because its attribution patterns are uniquely oscillatory across the context window: the proportion of important words rises and falls at regular intervals. Importantly, these oscillations are anti-phased across tasks (BA peaks where NWP dips, and vice-versa), reducing overlap.
>
> We conducted several checks (reported in Appendix): the oscillations persist under a different attribution method (IG), disappear for the same model on a different dataset (MRH) or using a smaller context length, and do not appear in Qwen2-1.5B despite shared architectural features (RoPE, GQA, FlashAttention2). These results suggest that the oscillations are stimulus- and context-dependent, not an artifact of our pipeline or of Llama’s architecture.
>
> # Missing masking experiments
> We thank the reviewer for this thoughtful comment. In the revised manuscript, we expanded Appendix C by adding POS-controlled masking, in which top-attributed words are replaced not with arbitrary random tokens (as in our original setup) but with random words sharing the same POS tag. This preserves local syntactic structure while selectively disrupting semantic content, providing a cleaner causal test. Because our attribution analyses show that BA-important words are mostly semantic or discourse-related, POS-based masking effectively acts as a semantic ablation: syntax remains intact, but meaning does not. Under this manipulation, BA performance still degrades, confirming that the semantic information carried by these words, not their syntactic role, is mostly what the BA decoder relies on.
>
> # Unclear “discourse” category
> We clarified the definition of the “discourse” category in Appendix A1. It captures story-specific, high-level content, such as character names, actions, or emotions, that mark the progression of the narrative. These features are used solely for interpretability and do not impose any constraints on the attribution method.
>
> # Necessity to test on more diverse datasets
> We thank the reviewer for raising this important point. While HP is our primary dataset, we intentionally included Moth Radio Hour (MRH) as a second, contrasting domain. MRH is a transcription of spoken, autobiographical storytelling, with different lexical statistics, discourse structure, and narrative pacing. Despite these differences, all core attribution trends replicate, suggesting that our findings are not tied to HP’s literary style.
>
> We agree that extending to non-narrative or multilingual data would be valuable. Due to time constraints we cannot include new datasets in the revision. However, as highlighted by the reviewer, our contribution is a general attribution pipeline that can be applied directly to such datasets. We now emphasize in the Future Work that exploring non-narrative and multilingual settings is an exciting direction enabled by the framework we introduce.
>
> # Clarity and presentation
> We appreciate the reviewer’s feedback and fully agree that clear visualization is essential for interpreting attribution patterns. To improve the paper, it would be extremely helpful to understand which specific aspects the reviewer found unclear or hard to interpret. Also, concerning the paper length, are there particular figures or sections that should be removed, condensed, or redesigned? We are committed to revising the figures, and the reviewer’s guidance on what was most confusing will help us prioritize the clearest and most informative presentation.

---

> > ### Author Response · Authors · 2025-11-24
> > **Answers to Questions**
> >
> > # Discussion on brain alignment and architectural factors
> > Llama3.2-1B achieves the highest brain alignment overall, showing the strongest mean voxel-wise correlation across layers and subjects. Falcon3-1B performs the worst, with significantly lower alignment than all other models except Mamba-1.4B. These differences are statistically supported by pairwise significance tests.
> > Importantly, Llama3.2-1B’s advantage does not come from parameter count but from representational quality. It inherits the Llama-3 training procedure, high-quality data curation, tokenizer, and RoPE positional encoding, that are known to enhance semantic coherence even at small scale. Consistent with this, our attribution analyses reveal that Llama3.2-1B uses broader and more structured contextual information than the other models, which likely contributes to its stronger alignment with brain activity.
> >
> > While we cannot draw causal conclusions about architectural drivers from these comparisons alone, our findings suggest that specific training and architectural features, such as robust long-range context handling or normalization strategies, may be responsible for the higher brain alignment of Llama3.2-1B. Future work can directly test these hypotheses by using our attribution pipeline to ablate individual components (e.g., RoPE, attention configuration, or normalization layers) and measure how brain alignment and attribution profiles change. This would allow us to more precisely identify which mechanisms contribute most strongly to brain-relevant representations.
> >
> > # Possibility to do more granular feature-based analysis
> > Thank you for pointing out this interesting possibility. The Harry Potter dataset indeed contains rich semantic annotations, and in principle one could define more fine-grained semantic subtypes. We chose to group words into three broad categories (syntactic, semantic, and discourse) because our goal was to capture the major functional distinctions that matter for comparing BA and NWP, while keeping the analysis stable and interpretable across models, subjects, and datasets. However, as appreciated by the reviewer, our pipeline is fully general, and nothing prevents applying it to finer semantic categories. We will add the extended feature-based analysis as promising future work.
> >
> > # Inter-subject stability of attribution patterns
> > We thank the reviewer for the thoughtful question. In Appendix F, we report per-subject attribution distributions to show that attribution patterns remain stable and consistent across subjects.
> >
> > # Adaptability to multimodal stimuli
> > We thank the reviewer for this insightful question. Tokenization is not a requirement for attribution methods. For instance, in the case of vision, input attribution maps generally consist of pixel-wise importance scores that highlight the importance of different regions in the input images. The gradient-based methods we used in this study can be applied to any differentiable mapping from stimulus features to outputs, and the overall attribution pipeline remains unchanged: we take the model’s stimulus-to-brain prediction, backpropagate gradients through the corresponding embedding layer, and analyze importance over these multimodal units exactly as we do for text tokens.

---

> > > ### Comment · Reviewer_P1Ga · 2025-11-27
> > >
> > > Thank you for the detailed response and for running extra experiments during this short rebuttal window.
> > > First, it's too bad the changes in the manuscript are not highlighted in a different color. It really helps track what was updated.
> > >
> > > Thanks for clarifying the Llama outlier behavior, the discourse category issue and the inter-subject stability of attribution patterns! These points are genuinely interesting. I also appreciate the clarification regarding the multimodal stimuli.
> > >
> > > You have added the POS-controlled masking experiment. This was an important point in my opinion.
> > >
> > > However, you can't claim in your answers for a general attribution pipeline with only evaluating it on two datasets. Either adjust how the contribution is framed or provide support across more datasets. It's especially what is valuable in your approach and you are only partially showing it in your current work.
> > >
> > > Regarding clarity and presentation: the figures repeated throughout the paper are not clearly explained. Both axes need to be explicitly described to be able to understand what we are looking at. The "top-% attribution" is a key concept in the paper yet the current explanation makes it hard to follow. You also repeat the  “distance from most recent word vs. proportion of important words” plot many times. Information can be condensed much more effectively. Page 22 to 24 could be merged into a single compact figure. It's in the appendix so it's not dramatic but for the main figures I would add more contextual information so the reader can directly understand.

---

> > ### Author Response · Authors · 2025-11-28
> >
> > We thank the reviewer for their constructive feedback and for carefully revisiting our revised analyses. We appreciate the positive comments regarding the clarification of Llama’s behavior, the discourse category, the multimodal applicability, and the new POS-controlled masking experiments.
> >
> > ## On highlighting manuscript changes.
> > We thank the reviewer for pointing this out. We misunderstood the guidelines on ICLR’s website and thought the pdfdiff was automatic.  We are sorry for the inconvenience, and we have now added a new revised version of the manuscript, containing color-highlighted changes.
> >
> > ## On the scope of the “general attribution pipeline.”
> > We thank the reviewer for raising this important point. Our intent is not to claim empirical generality across all stimulus modalities or linguistic domains, but rather to emphasize that the methodological pipeline itself is general and can be directly applied to other datasets. To avoid any overstatement, we have revised the framing throughout the manuscript, including the abstract (lines 17–18), introduction (lines 68–69), and discussion (lines 534–536, 549–553), to explicitly distinguish between (i) the generality of the method and (ii) the empirical scope of the present experiments. We now clearly state that our findings are demonstrated on two naturalistic fMRI datasets and do not claim broader empirical generality beyond them.
> >
> > ## On clarity and figure presentation.
> > We appreciate the reviewer’s constructive feedback regarding figure clarity. In response, we have substantially revised the captions of all main-text figures to explicitly describe the axes, the meaning of “top-% attribution,” and how proportions are computed. These clarifications ensure that readers can interpret the plots directly without needing to refer back to earlier sections.
> > Regarding the Appendix, we considered merging plots as suggested; however, we elected to retain per-model visualizations because they reveal model-specific positional patterns, particularly relevant for interpreting outlier behavior such as Llama3.2-1B’s oscillatory structure. We hope the reviewer finds the updated captions improve clarity while preserving the interpretive value of the detailed plots.
> >
> > We again thank the reviewer for the highly actionable suggestions, which have substantially improved the clarity, precision, and framing of the manuscript. We believe the revised version more accurately reflects the intended scope of the contributions.

---

### Official Review · Reviewer_Yxq3 · 2025-10-31

**Soundness:** 3
**Presentation:** 2
**Contribution:** 1
**Rating:** 2
**Confidence:** 4

**Summary:**

The paper evaluates the alignment between LLMs and human brains (primarily Harry Potter fMRI dataset) with respect to word features, for which the authors use a gradient-based input attribution method.
Specifically, input properties for brain alignment are compared with those for next-word prediction.
The major claim is that brain alignment and next-word prediction rely on distinct words: syntactic vs semantic information.

**Strengths:**

Claims are tested on several models and two fMRI datasets (primarily Harry Potter; Moth Radio Hour in Appendix).

Code available (currently in supplemental zip; GitHub promised after review).

**Weaknesses:**

### 1. Contribution
I am a bit lost with what to actually take from this paper. The methods themselves have previously been developed (brain encoding models as well as input attribution), and I'm not really sure why the claim of distinct word sets is important. I'm not really convinced that different words being deemed "important" (i.e., in a very fine-grain perspective) maps onto a fundamental difference in what NWP optimizes for and what brains optimize for. We know from the broad success of LLMs that NWP yields semantically very meaningful representations and capabilities, how does that reconcile with the claims here?

### 2. Causal relevance of word importance
The input attribution method estimates the importance of input words, which is causally quantified in Appendix C (good!). But I cannot tell from this analysis if the top% selected words are actually more important than e.g. randomly selected words.

I will note here that I find it a bit much to claim the approach the "introduc[tion of] a fine-grained input attribution method" (L013) for importing the Captum library and using a differentiable linear layer for projecting model features to brain activity (again in the conclusion L477 "We introduce the first end-to-end attribution framework for brain-LLM alignment"). Not every paper needs to claim a new method.



### Minor:
* It is at times difficult to follow the text with the amount of abbreviations used throughout (BA, NWP, TR, HP, IoU, ...).
* there's some inconsistency with the NWP<>BA studies between lines 086 and 051 where the sets of references are different. Hosseini et al. 2024 also seems relevant.
* After going through the paper several times (including the appendices), I just cannot find how NWP is actually measured. What dataset is used here?

**Questions:**

Why is the importance of words important?

Please addresses weaknesses 1 and 2 above.

---

> ### Author Response · Authors · 2025-11-24
>
> # Why is the importance of words important?
> We thank the reviewer for this conceptual question, which helped improve the framing of the paper. Our goal is not to claim that NWP lacks semantic structure, NWP is known to yield rich semantic representations. Instead, our contribution is to identify which parts of the input an LLM functionally uses when predicting brain activity, and how this differs from the information it uses for NWP. Prior work has studied linguistic drivers of brain–LLM alignment mainly via perturbing hidden representations or the input itself (Toneva et al., 2022; Oota et al., 2023b; Merlin & Toneva, 2024; Kauf et al., 2023). Our framework complements these approaches by providing word-level attributions, enabling analysis of not only what information matters but where in the context it resides. This positional dimension reveals that even if semantic information is broadly present in LLM representations, different tasks access it differently. NWP relies more on short-range syntactic cues (Oh & Schuler, 2023; AlKhamissi et al., 2025), whereas BA draws more on semantic and discourse content distributed broadly across the context. Thus, differences in “important word sets’’ reflect differences in task-specific usage of shared representations, not differences in representational richness.
>
> We will revise lines 39–64 to emphasize that our contribution is to characterize how BA and NWP functionally extract information from the same underlying representations, rather than merely identifying different sets of important words.
>
> # Causal relevance of word importance
> We thank the reviewer for this helpful comment. In the revised manuscript, we expanded Appendix C with two additional masking controls. We currently show preliminary results on Falcon3-1B and Gemma-2B, and will post a new comment as the complete set of results becomes available.
>
> We added: (1) a POS-matched random masking baseline, where we replace the same number of randomly selected words with random words of the same POS tag, and (2). a POS-controlled top-attributed masking condition in which top-attributed words are replaced by random words of the same POS tag. POS-based replacements ensure that any observed drop in performance cannot be attributed solely to syntactic violations or degenerate inputs.
>
> For NWP, top-attributed masking, whether random or POS-controlled, causes large increases in CE loss, whereas POS-matched random masking causes only minimal degradation. This confirms that the targeted words are genuinely more causally important than random words. For BA, performance drops substantially under all masking types, reflecting its broad semantic dependence. However, top-attributed masking (random or same-POS) consistently matches or exceeds the random baseline, showing that attribution still identifies above-baseline informative words even when syntactic coherence is preserved.
>
> These expanded experiments clarify the causal role of top-attributed words.
>
> # Overclaimed novel framework
> We appreciate the opportunity to clarify. We do not claim to introduce new attribution methods; we build on established techniques, for which we cite the appropriate references. Our contribution is providing a pipeline for applying attribution to brain-LLM alignment, enabling end-to-end word-level explanations and task-matched comparisons between BA and NWP using identical contexts and models. To avoid overstating novelty, we revised:
> - Line 013: “introduce a fine-grained input attribution method” → “describe a pipeline to apply attribution methods to the brain–LLM alignment setting”
> - Line 477: “introduce the first end-to-end attribution framework for brain-LLM alignment” → “apply attribution methods to brain-LLM alignment and analyze the resulting attribution patterns.”
>
> These revisions clarify that our novelty lies in the application and integration of existing attribution tools within the brain-encoding pipeline, along with the quantitative metrics we introduce to analyze attribution structure.
>
> # Use of abbreviations
> We thank the reviewer for this useful feedback. We removed several infrequently used abbreviations (HP, MRH, GXI, IG) to improve clarity.
>
> # Citation inconsistencies
> We thank the reviewer for pointing out this inconsistency. We fixed the inconsistent reference lists at lines 051 and 086 and added the missing Hosseini et al. (2024) citation.
>
> # Doubts on how NWP is measured
> We thank the reviewer for pointing this out. NWP is computed on the same text used in the brain experiments, ensuring that BA and NWP attributions are computed over identical input contexts. For each extended context window, we run the frozen LLM to obtain next-word logits. Attributions are then computed with respect to the cross-entropy loss  between the true and predicted tokens. No external dataset is used. Section 3.3.2 has been revised to make this point clearer.

---

### Official Review · Reviewer_3CbV · 2025-11-01

**Soundness:** 3
**Presentation:** 3
**Contribution:** 3
**Rating:** 6
**Confidence:** 2

**Summary:**

This paper presents a novel end-to-end attribution framework aimed at investigating the alignment between Large Language Models (LLMs) and human brain activity during language processing—referred to as Brain Alignment (BA). The framework is further applied to the debated question of the relationship between BA and Next Word Prediction (NWP). Overall, the work offers a valuable analytical tool and provides fresh insights into the types of information that BA and NWP respectively exploit.

**Strengths:**

1. The paper puts forward an "end-to-end" attribution framework that enables direct interpretation of the brain prediction model, advancing beyond the traditional representational correlation analyses found in previous research.
2. The authors conduct comprehensive cross-validation, examining various LLM architectures, two distinct fMRI datasets, and multiple evaluation metrics, which enhances the robustness of their results.

**Weaknesses:**

1.  The findings may primarily reflect what a linear decoder can extract from the LLM representations.  If a more powerful, non-linear encoder, such as a multilayer perceptron (MLP), were employed, the attribution outcomes could be substantially different.  The authors attribute their findings to the nature of the BA task, overlooking the strong inductive bias introduced by their chosen encoding model.
2.  Although Integrated Gradients (IG) are used for verification, this validation step is limited: IG is applied to only two representative models and is mainly used for feature and positional analysis.  However, the paper’s key claims—such as the Intersection over Union (IoU) analysis and attribution spread—are based entirely on Gradient x Input (GXI), making the justification potentially inadequate.
3.  The core linguistic feature analysis is performed solely on the HP dataset, which consists of data from just eight subjects reading one chapter.  Drawing broad conclusions—such as NWP favoring syntax and BA favoring semantics—based on this limited dataset is questionable in terms of generalizability.
4.  The study observes that Llama32-1B demonstrates a unique "oscillatory" attribution pattern on the HP dataset, but this pattern disappears in the MRH dataset and in short text contexts (80 words).  The authors suggest this may be "stimulus and context-dependent,” but this notable phenomenon is not thoroughly explained.

**Questions:**

see weakness.

---

> ### Author Response · Authors · 2025-11-24
>
> # Results may reflect inductive biases of the encoding models
> We appreciate the reviewer’s careful concern about inductive bias from the encoder. The goal of our attribution framework is to explain why brain–LLM alignment arises. For this reason, we deliberately used a linear encoder, which is the established standard in brain–language modeling (e.g., Schrimpf et al., 2021; Toneva & Wehbe, 2019). A linear mapping provides interpretable weights that reflect what information in the model’s representations mostly contribute to the predicted brain activity. In this way, attributions can be traced directly back to the input representations, ensuring that the resulting gradients reflect representational alignment between the LLM and the brain rather than additional nonlinear transformations learned by the encoder.
>
> # Missing results with Integrated Gradients
> Thank you for pointing this out, we agree that a fuller IG report will improve completeness. Overall, all results confirm the patterns highlighted by GXI analyses: low BA-NWP overlap, and greater NWP spread in early layers and greater BA spread in middle/late layers. In the revision, we will include the full IG results for Llama3.2-1B and Gemma-2B for IoU (Appendix E.1) and spread (Appendix E.2) to parallel the main analyses. Additionally, we are running the full attribution pipeline for all models using SmoothGrad. These results will be incorporated into the revised manuscript within the next few days.
>
> # Linguistic feature analysis performed just on the HP dataset
> We thank the reviewer for highlighting this limitation and agree that richer annotated datasets would deepen the feature-level conclusions. At present, HP is the only public fMRI dataset with detailed word-level annotations. We therefore used it as a proof-of-concept for the feature analysis: our goal was to illustrate how our attribution framework can incorporate and quantify such feature-level information when available.
>
> To address concerns about generalizability, we explicitly validated all non-linguistic analyses (IoU overlap, attribution spread, positional biases) on the Moth Radio Hour (MRH) dataset, which contains a different set of narratives, subjects, and recording conditions. The same qualitative patterns, such as distinct attribution distributions for BA and NWP, were observed across datasets and architectures, suggesting that our main conclusions do not depend on the HP stimulus itself.
>
> # Limited explanation of LLama’s oscillatory pattern
> We appreciate the reviewer’s concern, and agree that the oscillatory attribution pattern observed for Llama3.2-1B on the Harry Potter dataset requires clearer discussion. Importantly, our additional analyses show that:
> - Cross-method consistency: The oscillations persist when using Integrated Gradients, indicating they are not an artifact of our gradient-based pipeline.
> - Dataset dependence: The same model shows no oscillations on the Moth Radio Hour dataset, despite identical analysis parameters.
> - Context-length dependence: Reducing the context window to 80 tokens removes the oscillatory pattern entirely.
> - Architectural control: A closely related model (Qwen2-1.5B), which shares RoPE, GQA, and FlashAttention2, does not exhibit oscillations.
>
> Taken together, these findings suggest that the oscillatory effect arises from the interaction between the model and specific structural properties of the HP stimulus, rather than from the model architecture alone. While fully explaining the phenomenon is beyond the scope of the current work, we can offer a hypothesis grounded in the data: the HP text contains highly repetitive narrative scaffolding and regularly structured contextual cues (character names, dialogue markers, chapter transitions), which may induce an alternation of high and low attribution words when interacting with Llama’s strong long-range positional encoding and attention compression patterns. When those regularities disappear (as in MRH) or when the context window is too short to expose them, the oscillations vanish.
>
> We added a dedicated paragraph in the Discussion section of the revised manuscript, and we frame the observation of stimulus-models interactions as an interesting direction for future follow-up, for example by probing interactions between text structure and attribution dynamics.

---

> > ### Author Response · Authors · 2025-11-28
> >
> > Dear Reviewer 3CbV,
> >
> > We truly appreciate your effort in evaluating our work and your thoughtful suggestions.
> >
> > In response to your insightful comments, we have addressed the issues you highlighted and added new experiments. We uploaded a new revision of the manuscript, with all modifications clearly marked in color to help identify the updates. We believe these revisions significantly contribute to the clarity and completeness of the paper. In the following, we highlight some updates with respect to our previous comment.
> >
> > ## Update: Added preliminary results using SmoothGrad
> > Thank you again for the helpful suggestion to broaden the attribution analyses. As an update, we have now completed SmoothGrad attribution runs for all models except Zamba2-1.2B, and the results fully support the patterns reported with the other attribution methods: BA–NWP overlap remains low across thresholds, layer-wise spread differences replicate robustly, BA continues to exhibit broader, semantically driven contextual integration,
> >
> > We now updated:
> > - Appendix A.3 to explain how SmoothGrad works.
> > - Appendix E to add results with SmoothGrad on Llama3.2-1B, Gemma-2B, Falcon3-1B, and Mamba-1.4B. We will update them again as soon as Zamba2-1.2B’s run finishes.
> > - The main text (introduction to 3.3 and results section), to also reference these additional results as further confirmation that our findings are stable across multiple attribution methods.

---

### Official Review · Reviewer_UMmr · 2025-11-01

**Soundness:** 3
**Presentation:** 4
**Contribution:** 3
**Rating:** 6
**Confidence:** 4

**Summary:**

The manuscript presents a gradient-based method for computing token-level importance for brain alignment (BA) using a frozen LLM and gradients of the BA loss. Findings are compared to attribution analyses performed for LLM next word prediction. The relative similarities and differences between these attributions are studied.

**Strengths:**

The manuscript presents a method for computing token-level importance for brain alignment (BA) using a frozen LLM and gradients of the BA loss. The architecture is sound; the implementation used to backpropagate the BA objective is reasonable; the comparison to next-word prediction (NWP) attributions is rigorous.

**Weaknesses:**

Regarding masking, I think the current masking procedure is too strong, perhaps to the point that invalidates the procedure and conclusions drawn: it replaces the top words with random words and shows this removes brain prediction, but in doing so,  it also produces an input string that is not part of the training distribution, so the drop in performance cannot be interpreted as being solely due to removal of related information, but due to the introduction of what is nonsense content with respect to the LLM or encoder.  To interpret masking properly, a more sensitive procedure should be used, which removes the target-word information, but keeps the input phrase consistent with statistical distribution the model was train on.

A second point concerns within-method reliability. Although the LLM is frozen, both the BA encoder and the projection head (used to pass gradients back to tokens) can vary across splits and seeds. I suggest reporting seed-wise stability of BA attributions (e.g., seeded refits analyzed using within-BA IoU curves computed across seeds of *same* model). This is an internal ceiling against which BA-to-NWP intersections can be evaluated.

Finally, typical BA performance is modest (e.g., Pearson’s r ≈ 0.06 for predicted vs. observed activity). This corresponds to variance explained (R^2 = 0.0036), which is about 0.36%, i.e., **≈99.64%** of variance in brain activity  is unexplained. It is of course valid to compute gradients of a large loss and, as shown here, identify words whose masking eliminates even this small correlation . Still, attribution is most informative when model performance exceeds a practical threshold, or at least a threshold that is meaningful given prior work. To this end, I recommend reporting noise ceilings (e.g., from inter-subject reliability; often in range of 0.2-0.4 in other studies) and normalizing against them so the values in appendix G can be better understood.

**Questions:**

Suggestion: The conclusion that NWP and BA rely on different information is supported by the attribution content. The authors could also easily run a cross-task masking, where you mask BA-ranked words and measure NWP degradation, and vice versa. At present, masking is reported within task only.

**Details Of Ethics Concerns:**

The method offers a more detailed view of 'mind reading'. Because it can produce subject-specific word-importance maps, these not only constitute the typical 'fingerprint' that can be extracted from raw fMRI signal, but a potential view on content preferences by the individual. This is something the authors could address.

---

> ### Author Response · Authors · 2025-11-24
>
> # Concerns regarding the masking procedure
> We thank the reviewer for this helpful suggestion to strengthen the causal interpretation of the masking results. In the revised manuscript, we have expanded Appendix C to include **more sensitive and statistically controlled masking procedures**. We currently show preliminary results on Falcon3-1B and Gemma-2B, and will post a new comment as the complete set of results become available.
>
> In particular, we now introduce a **POS-controlled masking** method in which each top-attributed word is replaced with a random word from the same chapter sharing the same POS tag. This preserves local syntactic structure and avoids producing ungrammatical or distributionally atypical inputs, while still selectively removing semantic content. We also add a **POS-matched random masking baseline**, which replaces the same number of randomly selected words with same-POS alternatives. This allows us to isolate the effect of removing specifically the high-attribution words rather than words in general.
>
> For NWP, top-attributed masking, whether random or POS-controlled, causes large increases in CE loss, whereas POS-matched random masking causes only minimal degradation. This confirms that the targeted words are genuinely more causally important than random words, independent of syntactic disruptions.
> For BA, performance drops substantially under all masking types, reflecting its broad semantic dependence. However, top-attributed masking (random or same-POS) consistently matches or exceeds the random baseline, showing that attribution still identifies above-baseline informative words even when syntactic coherence is preserved.
>
> Together, these added experiments address the reviewer’s concern by ensuring that the masked inputs remain statistically plausible and by providing a controlled baseline against which causal effects can be interpreted.
>
> # Within-model reliability
> We thank the reviewer for raising this important point. We fully agree that assessing the stability of BA attributions across seeds provides a meaningful internal ceiling for interpreting BA–NWP intersections. We have already launched these multi-seed experiments for a subset of the models, but since this requires rerunning the full brain-encoding and attribution pipelines, which is computationally intensive, we may not be able to provide complete results in time for this rebuttal. If results complete during the rebuttal window, we will gladly post them in the thread.
>
> # BA scores normalization
> We thank the reviewer for the thoughtful observation, and we agree that reporting noise ceilings and normalized alignment scores will facilitate interpretation. We also realized that there was an error in our plotting script that led to lower mean raw correlations. Our per-ROI noise ceilings, computed using cross-subject prediction accuracy [1], are in the range 0.1- 0.15, in accordance with the values reported by [2,3]. The normalized correlation values are between 1 and 1.2 for all models, meaning the computed correlation scores are slightly above the noise ceiling. We updated Appendix G, by adding a plot showing the noise ceiling and updating the plots showing the normalized correlation values and the statistical tests.
>
> [1] Schrimpf M. et al. The neural architecture of language: Integrative modeling converges on predictive processing. Proceedings of the National Academy of Sciences, 118(45):e2105646118, November 2021.
> [2] Oota S. R., et al. What aspects of NLP models and brain datasets affect brain-NLP alignment?. CCN 2023
> [3] Aw, K. L. et al. Instruction-tuning aligns llms to the human brain. COLM 2024.
>
> # Cross-masking experiments
> We thank the reviewer for this constructive idea, which we agree would provide a valuable complementary test. We are currently completing the expanded masking suite requested across reviews; contingent on remaining time, we will also add the cross-task masking results.
>
> # Ethics concerns
> We appreciate the reviewer’s suggestion and have updated the paper accordingly.
>
> While our method provides fine-grained attribution analyses, it does not enable reconstruction of subject-specific semantic content or personal preferences. The only subject-level visualizations we report are positional attribution distributions (i.e., how strongly each task relies on earlier vs. later words in the context). These distributions reflect general processing dynamics rather than semantic content and cannot be used to infer individual mental states or preferences.
>
> Furthermore, all datasets we use are publicly released and fully anonymized. In the manuscript revision, we added an “Ethics statement” clarifying that (1) we do not release subject-identifiable attribution maps, (2) we only report aggregated linguistic and positional statistics, and (3) any future application of such fine-grained alignment methods should consider privacy-preserving data handling, explicit consent, and appropriate governance for brain-data usage.

---

> > ### Author Response · Authors · 2025-11-28
> >
> > Dear Reviewer UMmr,
> >
> > We truly appreciate your effort in evaluating our work and your thoughtful suggestions.
> >
> > In response to your insightful comments, we have addressed the issues you highlighted and added new experiments. We uploaded a new revision of the manuscript, with all modifications clearly marked in color to help identify the updates. We believe these revisions significantly contribute to the clarity and completeness of the paper. In the following, we highlight some updates with respect to our previous comment.
> >
> > ## Update: Within-model reliability
> > Thank you again for raising the important point regarding within-model reliability as an internal ceiling for interpreting BA–NWP IoU. As promised, we have now completed preliminary multi-seed experiments for three models (Falcon3-1B, Llama3.2-1B, Gemma-2B), using three randomly generated seeds (865, 395, 777). The full results for all models are currently running; once they complete during the rebuttal period, we will update this thread.
> > Below we report the within-BA IoU across seeds for several attribution-mass thresholds. These values quantify how similar the BA top-attributed words are across seed-wise refits of the brain encoder, and thus provide an interpretable upper bound on BA–NWP IoU comparisons.
> >
> >
> > | **Model**        | **IoU @1%**       | **IoU @5%**       | **IoU @10%**      | **IoU @60%**      | **IoU @80%**      |
> > |------------------|--------------------|--------------------|--------------------|--------------------|--------------------|
> > | **Falcon3-1B**   | 0.994 ± 0.080      | 0.991 ± 0.072      | 0.989 ± 0.062      | 0.992 ± 0.011      | 0.994 ± 0.007      |
> > | **Llama3.2-1B**  | 0.990 ± 0.099      | 0.985 ± 0.087      | 0.982 ± 0.071      | 0.973 ± 0.018      | 0.977 ± 0.013      |
> > | **Gemma-2B**     | 0.992 ± 0.088      | 0.990 ± 0.079      | 0.989 ± 0.065      | 0.992 ± 0.012      | 0.993 ± 0.007      |
> >
> >
> > Across all examined models and thresholds, within-BA IoU is extremely high (0.97–0.99+), indicating that: (1) BA attributions are highly stable across seeds, even with a fully refit brain-encoding model, and (2) the BA–NWP IoU values reported in the main text (~0.05–0.10 for low thresholds) are far below this internal ceiling, meaning that the low cross-task overlap cannot be attributed to instability in BA attribution.

---

### Author Response · Authors · 2025-12-02
**Rebuttal Summary**

We thank the reviewers for their careful feedback. Below is a brief summary of the revisions and additional analyses we performed.
- **Conceptual framing and novelty (Yxq3, 4jvo)**: We clarified throughout the abstract, introduction, and discussion that our main contribution is a pipeline that applies established attribution methods to brain–LLM alignment, rather than introducing new attribution techniques. The framework enables end-to-end, word-level explanations and BA–NWP comparisons on identical contexts, characterizing how the two tasks functionally extract information from the same underlying representations (which linguistic information, where in the context, and how broadly).
- **Why “important words” matter (Yxq3)**: We position word-level attributions as complementary to prior perturbation work on linguistic drivers of BA. Our analysis asks not only what information matters but where it is located in the context. We emphasize that differences in “important word sets” reflect task-specific usage of shared representations, not a lack of semantic richness in NWP. We revised the introduction accordingly.
- **Within-model reliability (UMmr)**: We ran the full attribution pipeline with three random seeds for three models (Falcon3-1B, Llama3.2-1B, Gemma-2B). Within-model BA IoU is extremely high across thresholds and far above the BA–NWP IoU values, showing that low BA–NWP overlap cannot be explained by unstable BA attributions. Results for the remaining models will be included in the camera-ready version.
- **Masking experiments (UMmr, Yxq3, P1Ga)**: We expanded Appendix C with (1) POS-controlled masking of top-attributed words (replacing each with a random same-POS word from the same chapter) and (2) a POS-matched random masking baseline with the same number of masked words. These procedures preserve local syntax and show that masking top-attributed words affects both NWP and BA more than masking random words, supporting their causal relevance.
- **Incompleteness of CoM metric (4jvo)**: We added an attribution-spread measure (weighted positional standard deviation) to quantify whether attributions are tightly clustered (strong recency/primacy) or broadly distributed, and we updated all relevant figures accordingly.
- **Additional attribution methods (3CbV, 4jvo)**: We added full Integrated Gradients analyses (IoU, spread, feature-based, positional) for Llama3.2-1B and Gemma-2B, and SmoothGrad results for four of five models, described in Appendix A.3 and reported in Appendix E. All methods consistently show low BA–NWP overlap, greater NWP spread in early layers, and broader, semantically driven BA integration. The remaining SmoothGrad run (Zamba2-1.2B) is in progress and will be included in the camera-ready version.
- **Brain-alignment scores and encoder concerns (UMmr, 3CbV)**: We identified and fixed a plotting bug that underestimated raw correlations. We now report noise ceilings and normalized BA scores: the resulting values are consistent with prior work. We also clarify our choice of a linear encoder, following standard brain–language practice, to limit inductive bias and keep the mapping from model representations to brain activity interpretable.
- **Task definition and additional analyses (Yxq3, P1Ga)**: We clarify that NWP is computed using the same input text as BA: no external dataset is used. We expanded the discussion on model classes and Llama3.2-1B’s oscillatory behavior, clarified the definition of the “discourse” category, and summarized inter-subject stability of attribution patterns. We note that more fine-grained feature analyses and extensions to multimodal datasets are natural future applications of the pipeline.
- **Datasets and generalizability (P1Ga)**: We highlight that Harry Potter and Moth Radio Hour differ substantially in lexical statistics, discourse structure, and narrative style, yet all core attribution trends replicate across them. To avoid overstatement, we revised the abstract, introduction, and discussion to distinguish the generality of the attribution pipeline from the empirical scope of the present experiments and to frame multimodal/non-narrative datasets as future work.
- **Ethics (UMmr)**: We added an ethics statement clarifying that we use public, anonymized data, and that our pipeline does not enable reconstruction of subject-specific semantic content or personal preferences, while emphasizing privacy-preserving practices for future applications.
- **Presentation improvements (P1Ga, Yxq3)**: We substantially revised the captions of all main-text figures (axes, “top-% attribution,” proportion definitions), cleaned up abbreviations, and fixed citation inconsistencies.

---

### Meta-Review · Area_Chair_NRhi · 2026-01-06

**Summary:**

The authors proposed a new method to compare how LLMs align with brain activities (BA) vs next-word prediction (NWP) through the token-level importance (end-to-end attributions). The authors compare multiple LLMs on two datasets and provide insights on the differences between predictive and comprehension-related representations. This paper has mixed reviews -- some reviewers find the proposed framework useful while some reviewers questioned the limited novelty of this paper. Shared concerns include the limited evaluations (mostly on 2 datasets). While the authors provided clarifications in the rebuttal with additional experiments, some of the reviewers concerns are not fully addressed, including inadequate justification, large unexplained variance, limited benchmarks.

**Reviewer Concerns:**

Reviewer UMmr (6, conf 2)
- masking procedure too strong --> proposed POS-controlled masking
- method reliability, request to report seed-wise stability --> posted a table with 3 random seed; not all models available
- BA performance variance unexplained 99.64%, request to report noise ceiling --> results not available in rebuttal timeframe
- ethics concerns --> addressed

Reviewer 3CbV (6, conf 2)
- only linear encoder in BA is used, what about non-linear encoder? --> addressed, deliberately use linear encoder
- validation step is limited, justification inadequate (IG v.s. GXI) --> added smoothgrad attribution; but still did not directly address reviewer's question of IG v.s. GXI
- linguistic feature only analyzed on HP dataset, lack of generalizability --> clarified HP is the only public fMRI dataset with detailed word-level annotation; did experiment on MRH dataset instead
- not well-explained Llama32-1B on HP dataset --> added discussion + observation

Reviewer Yxq3 (2, conf 4)
- concerns on contribution --> clarified the contribution is to identify which parts of the input an LLM functionally uses when predicting brain activity, and how this differs from the information it uses for NWP
- causal relevance of word importance --> expanded App. C with two masking controls

Reviewer P1Ga (4, conf 2)
- under-developed discussion on model class --> addressed by observations summary
- attribution --> addressed
- dataset dependence, largely based on HP dataset and MRH dataset, want to see non-narrative or multi-lingual data --> reviewer not satisfied
- clarity and presentation --> partially addressed

Reviewer 4jvo (2, conf 4)
- limited conceptual novelty --> partially addressed with clarification that input attribution is used to explain brain-LLM alignment and quantitative metrics is adapted to analyze attribution patterns. The BA-NWP comparison serves as a case study illustrating the types of insights enabled by this framework.
- incompleteness of CoM metric --> addressed by a new complementary metric
- gradient-based attribution limitations, request smoothgrad result --> addressed by additional experiments

**Reviewer Scores:**

- Reviewer UMmr would remain the score of 6 or downgrade to 5; concerns are partially addressed, requested experiment of noise ceiling is not provided by the authors; requested experiment of method reliability is partially provided by the authors

- Reviewer 3CbV would remain the score of 6 or downgrade to 5; concerns are partially addressed, the question on inadequate justifications is not well-addressed, even though showing another method smoothgrad (not what the reviewer asked)

- Reviewer Yxq3 would increase score from 2 to 3; concerns on causal relevance of word importance is addressed; contribution and novelty is still a concern

- Reviewer P1Ga would increase score from 2 to 3; concerns are partially addressed; evaluation on limited (2) dataset is still a concern

- Reviewer 4jvo would likely increase score from 2 to 4; concerns are mostly addressed, the limited conceptual novelty may still be a concern

---

### Decision · Program_Chairs · 2026-01-26

Reject